# LESS IS MORE: EXPLOITING FEATURE DENSITY FOR ENHANCED MEMBERSHIP INFERENCE ATTACKS

## ABSTRACT

Membership inference attacks have become the de facto standard for assessing privacy breaches across various machine learning (ML) models. However, existing approaches often require substantial resources, including large numbers of shadow models and auxiliary datasets, to achieve high true positive rates (TPR) in the low false positive rate (FPR) region. This makes these attacks prohibitively expensive and less practical. In this work, we propose a novel membership inference attack that exploits feature density gaps by progressively removing features from both members and non-members and evaluating the corresponding model outputs as a new membership signal. Our method requires only a few dozen queries and does not rely on large auxiliary datasets or the training of numerous shadow models. Extensive evaluations on both classification and diffusion models demonstrate that our method significantly improves the TPR at low FPR across multiple scenarios.

## 1 INTRODUCTION

Enabled by the availability of extensive, high-quality datasets, the field of machine learning (ML) has seen remarkable progress (Hu et al., 2022), with ML-driven technologies increasingly integrated into critical societal operations (Maslej et al., 2023). However, since these datasets often contain sensitive personal information, such as medical records, it is essential to ensure that ML models do not compromise the confidentiality of their training data. Membership inference attacks (MIAs), which aim to determine whether a specific data point was included in a model's training set, have become the de facto standard for quantifying privacy leakage in various ML models (Shokri et al., 2017; Salem et al., 2018; Song & Mittal, 2021; Carlini et al., 2022; Liu et al., 2022a). Furthermore, the widespread use of public datasets raises legal concerns, particularly regarding the European General Data Protection Regulation (GDPR)'s *right to be forgotten* (Shastri et al., 2019). In this context, MIAs enable individuals to verify if ML service providers are using their personal data, allowing them to request its removal in compliance with GDPR regulations.

Most existing MIA studies directly use the model's output posteriors or derived metrics (*e.g.*, loss) to launch attacks (Yeom et al., 2018; Sablayrolles et al., 2019; Shokri et al., 2017; Salem et al., 2018; Song & Mittal, 2021). While effective on average-case metrics such as accuracy and AUC, these methods perform poorly in true-positive rate (TPR) at low false-positive rate (FPR), which is the de facto standard for evaluating MIA established by Carlini et al. (2022). This limitation arises because certain non-member samples with distinctive features can closely resemble members in terms of model output, leading to higher false-positive rates in these approaches. Recent techniques address this issue from three main perspectives. The first approach involves quantifying the difficulty of individual sample points and using this information to adjust the model's original outputs (Watson et al., 2022; Shi et al., 2024). This is typically done by calculating the membership score as the difference between the outputs of the target model and a reference model trained on auxiliary data drawn from the same distribution. The second approach learns per-sample hardness by training a large number (*e.g.*, several hundred) of shadow models (Carlini et al., 2022; Wen et al., 2023). These models are used to estimate the distribution of output logits for examples both inside and outside the training set. The third approach identifies new membership signals. Liu et al. (2022a) observe that easy non-members show consistently low losses during training, and they leverage an auxiliary dataset to perform knowledge distillation and use intermediate loss trajectories for membership inference. However, these approaches all require substantial resources, either auxiliary datasets drawn

from the same distribution as the target model or the training of a large number of shadow models. This makes membership inference attacks prohibitively expensive and less practical.

To reduce the required resources while maintaining attack effectiveness, we propose a scalable membership inference attack that significantly improves the TPR at low FPR. Our method requires only a few dozen queries and does not rely on large auxiliary datasets or the training of numerous shadow models. Leveraging the insight that member samples reside in high-density regions of the learned feature space, the model is expected to maintain higher confidence in these samples, even with partial feature removal. We progressively remove features from both groups and evaluate the model's predictions as the new membership signal. Extensive evaluations on both classification and diffusion models across various benchmark datasets, architectures, and adversarial settings demonstrate that our attack consistently outperforms state-of-the-art methods across multiple scenarios. Furthermore, a major advantage of our approach is its applicability to pre-trained models without the need to train shadow models, which makes our method particularly efficient when applied to large-scale models.

Our contributions are summarized as follows: (1) We introduce a novel feature removal strategy to identify a new membership signal. (2) Our approach is resource-efficient, eliminating the need to train numerous shadow models or use large auxiliary datasets. (3) We conduct extensive empirical validation, demonstrating that our approach significantly improves the TPR at low FPR and generalizes to both classification and diffusion models across various datasets and model architectures.

## 2 BACKGROUND AND RELATED WORK

Membership inference attacks (MIAs) aim to determine whether a particular record or data sample was part of the training dataset used for the ML model, which has been acknowledged as the de facto standard for evaluating ML models' privacy risks (Carlini et al., 2022; Liu et al., 2022a).

MIAs have been developed across different threat models with varying levels of adversarial knowledge. They can be performed in the white-box settings (Leino & Fredrikson, 2020; Nasr et al., 2019) in which the adversary has the knowledge of the model's architecture and parameters, but most of the attacks have been developed for more practical black-box settings (Shokri et al., 2017; Salem et al., 2018; Carlini et al., 2022; Liu et al., 2022a) in which the adversary only has the query access to the target ML model. Early MIAs exploit the insight that ML models, trained to minimize the loss of their training data, often exhibit generalization gaps between training and testing samples. These attacks directly leverage this performance disparity to infer membership status based on metrics such as loss (Yeom et al., 2018; Sablayrolles et al., 2019), confidence scores (Shokri et al., 2017; Salem et al., 2018), and entropy (Song & Mittal, 2021). On the other hand, Jayaraman et al. (2021) infers membership status by measuring changes in model loss when small random noise is added to inputs. While this method shares some similarities with our feature manipulation approach, there are key differences: First, our method emphasizes the contribution of individual features to the model's decision, recognizing that not all image regions equally affect the output (Selvaraju et al., 2017). In contrast, adding random noise indiscriminately may overlook these fine-grained disparities between member and non-member samples. Second, the effectiveness of a perturbation-based attack heavily relies on the perturbation accurately reflecting the importance of the features being perturbed, making any resulting changes in predictions directly attributable to these perturbations. In Jayaraman et al. (2021), random noise can lead to distribution shifts (Hooker et al., 2019) and adversarial artifacts (Fong & Vedaldi, 2017), obscuring the source of prediction changes and potentially confounding the analysis. While effective on average-case metrics like accuracy and AUC, these methods struggle to perform well at low false-positive rates (FPR), a de facto standard established by Carlini et al. (2022) for evaluating MIA.

Recent techniques tackle this issue from three main perspectives. The first line of approaches quantify the difficulty of sample points and use this value to adjust the model's original outputs (Shi et al., 2024; Watson et al., 2022). Shi et al. (2024) proposed a learning-based difficulty calibration (LDC) attack. Their method requires an auxiliary dataset in addition to shadow datasets to train a reference model, which is used to calibrate both the shadow model's and the target model's behavior on a data record by computing the loss difference between the respective models and the reference model. The second line of approach learns per-sample hardness by using statistical testing (Carlini et al., 2022; Wen et al., 2023). Carlini et al. (2022) trained a large number (*e.g.*, several hundred) of shadow models to learn the distribution of model output logits on examples in and out

of the training set. The third line of approaches identify new membership signals to differentiate members and non-members that exhibit similar low losses (Liu et al., 2022a). Liu et al. (2022a) exploited the model training process and observed that easy non-members exhibit consistently low losses. They assume the adversary possesses an auxiliary dataset to perform knowledge distillation, simulating the actual model training process, and use the obtained intermediate loss trajectories to facilitate membership inference. However, these approaches require substantial resources. The first and third approaches necessitate a large auxiliary dataset drawn from the same distribution as the target model's training set, in addition to the shadow datasets used for training shadow models. The third approach requires training a large number of shadow models, which can be computationally expensive, especially when attacking large-scale models such as diffusion models (Ko et al., 2023).

Similarly, generative models are also vulnerable to membership inference attacks (Hayes et al., 2019; Chen et al., 2020). Given diffusion models have recently surpassed GANs as the leading generative technique (Yang et al., 2023), they have become an emerging focus of MIA research (Wu et al., 2022; Matsumoto et al., 2023; Duan et al., 2023). Matsumoto et al. (2023) investigated MIAs against diffusion models in white-box settings, utilizing the models' loss values (*i.e.*, noise estimation errors) to infer membership status. Recently, Duan et al. (2023) proposed the Step-wise Error Comparing Membership Inference (SecMI) attack, achieving superior performance against diffusion models. The key insight of their approach is that member samples have smaller posterior estimation errors compared to non-member samples during the forward posterior estimation.

## 3 EXPLOITING FEATURE DENSITY

### 3.1 THREAT MODEL

We assume a commonly adopted black-box scenario in which the adversary has access only to the output posterior of the target model, without knowledge of its internal parameters. Additionally, we assume that the adversary is aware of the target model's architecture and possesses an auxiliary dataset that shares the same distribution as the target models' training dataset. This setting is commonly used for training shadow models in most existing works (Li et al., 2022; Salem et al., 2018; Song & Mittal, 2021; Li & Zhang, 2021; Carlini et al., 2022; Liu et al., 2024). During the experiment, we further explore scenarios where the adversary has access to a large supplementary dataset in addition to the shadow datasets as in Liu et al. (2022a) and Shi et al. (2024), and is capable of training a substantial number of shadow models as in Carlini et al. (2022). Furthermore, we investigate the attack effectiveness under more practical scenarios where the adversary lacks knowledge of the target model's architecture and training data distribution.

### 3.2 DESIGN INTUITION

To address the resource demands and improve performance at low FPRs, we aim to propose a new MIA signal that more effectively exploits the gaps between members and non-members.

Machine learning models are known to generalize better on examples similar to those on which they have been trained (Goodfellow, 2016). Previous membership inference attacks leverage the generalization gaps between training data and testing data, based on the insight that models behave differently on data they have seen during training compared to unseen data. In this paper, we extend this observation by demonstrating that these generalization gaps not only occur between exact copies of the training and testing data but also between similar variants. This phenomenon can be explained through the lens of feature

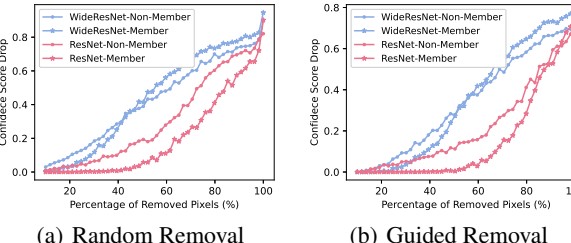

(a) Random Removal  (b) Guided Removal

Figure 1: The confidence score drops for members and non-members that have similar small ($< 0.01$) losses of different removal strategies on models trained on CIFAR-100.

learning theory (Cao et al., 2022; Kou et al., 2023), which suggests that a model's generalization

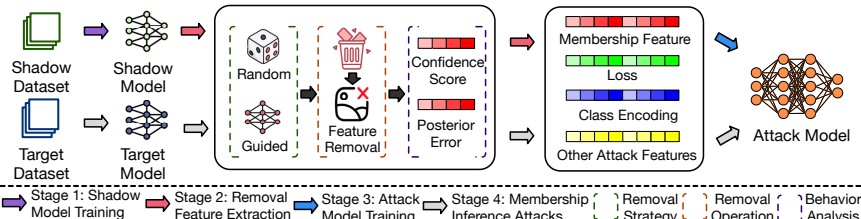

Figure 2: Overview of our proposed attack framework.

capability is directly related to the sample size and the signal-to-noise ratio of the features. During training, models are repeatedly exposed to member samples, causing these samples to lie in high-density regions of the learned feature space. In contrast, non-member samples are less likely to fall into these high-density regions, especially in complex models. More specifically, we define the density of the learned representation as follows: Let $\phi(x)$ represent the feature embedding of a sample $x$ in a model, then the density at $\phi(x)$ can be defined in terms of its proximity to neighboring samples: $p(\phi(x)) = \sum_{x_i \in N(x)} exp(-||\phi(x) - \phi(x_i)||^2)$, where $N(x)$ are the k-nearest neighbors of $\phi(x)$ in the dataset. Member samples exhibit higher $p(\phi(x))$ values than non-member samples because the model has learned more densely around these points. We empirically validate this, with details given in Appendix D. Based on this hypothesis, we propose that systematically removing or altering features from the input should affect the model's confidence differently for members and non-members. Specifically, the model's ability to make confident predictions should be more resilient for member samples than for non-member samples. This is because member samples, being close to other similar samples in the feature space, can still be recognized by the model even with partial feature removal, allowing it to predict confidently. To verify this hypothesis, we visualize the results for members and non-members exhibiting similarly small losses (*i.e.*, less than 0.01) using two different models trained on CIFAR-100 and applying two proposed feature removal strategies. In this case, loss alone is insufficient to distinguish between members and non-members. As shown in Figure 1, the confidence scores of members decrease more gradually, especially during the early stages of feature removal. Since we assume black-box threat model, the adversary cannot directly manipulate the intermediate feature representation. Instead, the adversary, having control over the input, can leverage pixel removal to approximate the feature removal process. This approach relies on the inherent functionality of neural networks, where input modifications (such as pixel removal) propagate through the network and result in changes to the intermediate feature representations. By simulating feature removal in this way, the adversary effectively influences the learned feature space, enabling the attack within the constraints of the threat model.

### 3.3 ATTACK METHOD

Building on this insight, we propose a novel membership inference attack centered around the feature removal scheme. An overview of our attack framework is provided in Figure 2, which contains four stages: shadow model training, removal feature extraction, attack model training, and membership inference attack.

**Shadow Model Training.** As mentioned in Section 3.1, the adversary has access to an auxiliary dataset $\mathcal{D}_{shadow}$ drawn from the same distribution as the training dataset of the target model. The adversary then splits the auxiliary dataset into two disjoint subsets: $\mathcal{D}_{shadow}^{train}$ and $\mathcal{D}_{shadow}^{test}$. The former subset is utilized to train the shadow model $\mathcal{M}^S$ and acts as the member samples, while the latter serves as non-member samples.

**Removal Feature Extraction.** During this stage, the adversary performs feature removal on $\mathcal{D}_{shadow}$ to extract features crucial for the attack. For a sample $x$, the adversary chooses a removal ratio and applies a removal strategy to select features and perform removal operations. We introduce two strategies: (1) Random-based Removal (Ours. Random.), which randomly removes a specified ratio of pixels from the input, and (2) Guided-based Removal (Ours. Guided.) ranks the predicted mask values from the mask prediction model in ascending order and removes features of $x$ whose mask values fall below a given percentile threshold, based on the specified removal ratio. The adversary then measures the model output changes (*e.g.*, confidence score changes in classification models, and forward posterior estimation errors in diffusion models). By varying the removal ratio,

a trajectory of output is obtained, which is regarded as the new membership feature. Further details on the feature removal operations, strategy, and behavior selection will be provided in Section 3.4.

**Attack Model Training.** The adversary aggregates the extracted shadow removal-based membership features, along with the loss computed by querying the shadow model $\mathcal{M}^S$ and the one-hot encoding of the true classes, as the final membership features. For diffusion models, we only use removal-based membership features as the final membership feature vector. In cases where the adversary has additional capabilities, the respective membership features can be further integrated within the feature set. Subsequently, the adversary trains an attack model on the membership features using a Multi-Layer Perceptron (MLP) network.

**Membership Inference Attack.** In the final stage of the MIA, the adversary applies the same feature removal process to the target sample. The resulting features, along with the loss, one-hot encoding of the classes, and any additional attack features (*e.g.*, concatenation with loss trajectories as described in Liu et al. (2022a) (Ours w/ loss traj.)), are input into the previously trained attack model to determine the membership status of the target sample.

### 3.4 FEATURE REMOVAL SCHEME

The design of our feature removal scheme is centered around our design intuition, that is, how to design a feature removal scheme that effectively exploits the density gaps between members and non-members. To this end, we address three key questions: (1) How are features removed (*e.g.*, setting them to default values)? (2) What strategy guides the removal of features (*e.g.*, sequential removal following a specific order)? (3) Which aspects of model behavior are analyzed following feature removal (*e.g.*, confidence score changes)? These elements collectively define our approach.

**Feature Removal Operation.** Ideally, feature removal operations should accurately reflect the importance of the removed features, so that the changes in the model's predictions can be directly attributed to the removal of those features. Feature removal has been widely studied in the context of explainable ML as a means to quantify the impact of individual features on a model's predictions (Fong & Vedaldi, 2017; Covert et al., 2021). A common method involves setting the removed pixels to a fixed value (*e.g.*, zero) (Petsiuk et al., 2018). However, such removal operations have been shown to induce distribution shifts (Hooker et al., 2019) and adversarial artifacts (Fong & Vedaldi, 2017), complicating the determination of whether prediction changes result from the feature removal itself or from the induced distribution shifts and artifacts. To address this challenge, we leverage Noisy Linear Imputation (Rong et al., 2022), which estimates the values of removed pixels using a weighted mean of their neighboring pixels. When multiple pixels are removed, this process forms a system of equations where known pixel values are used directly, while removed pixels are treated as unknown variables, resulting in a linear equation system, that is sparse and can be effectively solved. The strong correlation between a removed pixel and its neighbors mitigates the distribution shift, as the imputed values are consistent with the underlying distribution. Additionally, averaging neighbors inherently smooth these regions, which helps mitigate adversarial artifacts by reducing abrupt changes (Fong & Vedaldi, 2017).

**Feature Removal Strategy.** Images typically have high dimensionality (Deng et al., 2009), which poses challenges when removing pixels one by one for analysis, as this process would require a large number of queries. To address this, we randomly group pixels into clusters and then progressively remove an increasing percentage of these pixel groups. By doing so, we effectively reduce the dimensionality from a large number of pixels to a manageable number of clusters. However, this approach overlooks the varying importance of image features in decision-making (see Appendix H), limiting its ability to exploit the feature density gaps between members and non-members. To overcome this, we devise an approach to estimate feature importance by identifying the smallest region of the image that still allows for confident classification. This involves masking the image:

$$\hat{x} = x \odot m + r \odot (1 - m), \tag{1}$$

where $x$ is the original image, $m$ is the mask, $r$ represents the features replacing the removed areas, and $\odot$ means element-wise multiplication. $m$ has the same shape as $x$, with each element taking a continuous value between 0 and 1. We measure the impact of feature removal on model performance using the Carlini & Wagner (C&W) loss function (Carlini & Wagner, 2017):

$$CW(\hat{x}, t) = \max(\max_{i \neq t} \mathcal{M}^C(\hat{x})_i - \mathcal{M}^C(\hat{x})_t, -k), \tag{2}$$

where $\mathcal{M}^C$ represents the classifier model, $t$ represents the original class predicted by $\mathcal{M}^C$ before removal. We also need to restrict the area of the mask, so we take the $l_1$ norm of the mask to ensure the sparsity of the elements. The design of $r$ is important, as it could incur class information leakage and adversarial artifacts. Initially, we aimed to leverage noisy linear imputation to generate $r$. However, this operation requires a binary mask to clearly identify the removal pixels, making the optimization non-differentiable. To tackle this, we alternately use a Gaussian-blurred version of the original image and a random color image with added Gaussian noise, selected with a certain probability. This approach mitigates the class information leakage resulting from the removal operation while ensuring an end-to-end differentiable optimization. Furthermore, to mitigate adversarial artifacts, which generally arise from unnatural noises (Goodfellow et al., 2014; Fong & Vedaldi, 2017), we adopt the total variation penalty to regulate the mask $m$ to have a more natural and smooth shape:

$$TV(m) = \sum_{i,j} (m_{i,j} - m_{i,j+1})^2 + (m_{i,j} - m_{i+1,j})^2. \tag{3}$$

Consequently, our loss function is formulated as:

$$L(m) = CW(x \odot m + r \odot (1 - m), t) + \alpha \cdot TV(m) + \beta \cdot l_1(m). \tag{4}$$

We then train a mask prediction model, $\mathcal{P}$, using the above loss function. We adopt the U-net architecture (Ronneberger et al., 2015) for $\mathcal{P}$. The details of the models are given in Appendix B.

**Model Behavior Selection.** For classification models, we use the confidence score of the original class as the metric. For diffusion-based generative models, since they do not provide prediction results, we cannot obtain such confidence scores. Inspired by Duan et al. (2023), who showed that diffusion models are trained to match the forward process posterior distribution at each timestep, and thus members have smaller posterior estimation errors compared to non-members. We also leverage this posterior estimation error as the model behavior metric.

## 4 EXPERIMENTS

### 4.1 EXPERIMENTAL SETUP

**Datasets and Models.** For our experiments, we employ three benchmark datasets commonly used in membership inference attack studies: CIFAR-10, CIFAR-100 (Krizhevsky et al., 2009), and CINIC-10 (Darlow et al., 2018). Consistent with existing works (Salem et al., 2018; Yuan & Zhang, 2022; Li et al., 2022; Liu et al., 2022a), each dataset is divided into four equal subsets: $\mathcal{D}_{target}^{train}$, $\mathcal{D}_{target}^{test}$, $\mathcal{D}_{shadow}^{train}$, and $\mathcal{D}_{shadow}^{test}$. The $\mathcal{D}_{target}^{train}$ subset is used to train the target model $\mathcal{M}$, and $\mathcal{D}_{shadow}^{train}$ is utilized to train the shadow model $\mathcal{M}^S$. We mainly employ two widely used neural network architectures for our experiments, ResNet-18 (He et al., 2016) and WideResNet-32 (WRN-32) (Zagoruyko & Komodakis, 2016). Additionally, we further investigate VGG-16 (Simonyan & Zisserman, 2014) and DenseNet-161 (Huang et al., 2017) during the ablation study. For diffusion models, we employ the popular DDPM (Ho et al., 2020). To mitigate model overfitting, we employ standard training techniques such as weight decay (Krogh & Hertz, 1991) and train-time augmentations (Cubuk et al., 2018). Detailed dataset and model settings are provided in Appendix A.

**Evaluation Metrics.** In alignment with state-of-the-art studies (Liu et al., 2022a; Carlini et al., 2022), we primarily evaluate the attack performance at a low False Positive Rate (FPR). Specifically, we employ the following metrics: (1) Full Log-scale Receiver Operating Characteristic (ROC) Curve; (2) True Positive Rate (TPR) at Low False Positive Rate (FPR), which measures attack performance at a specific FPR (*e.g.*, 0.1%). For completeness, we also report average case metrics commonly used in membership inference attacks (Shokri et al., 2017; Song & Mittal, 2021; Salem et al., 2018), including: (3) Balanced Accuracy and Area Under the ROC Curve (AUC). To maintain

consistency, the balanced accuracy is calculated using a fixed threshold of 0.5 on the softmax output of MLP to determine membership status.

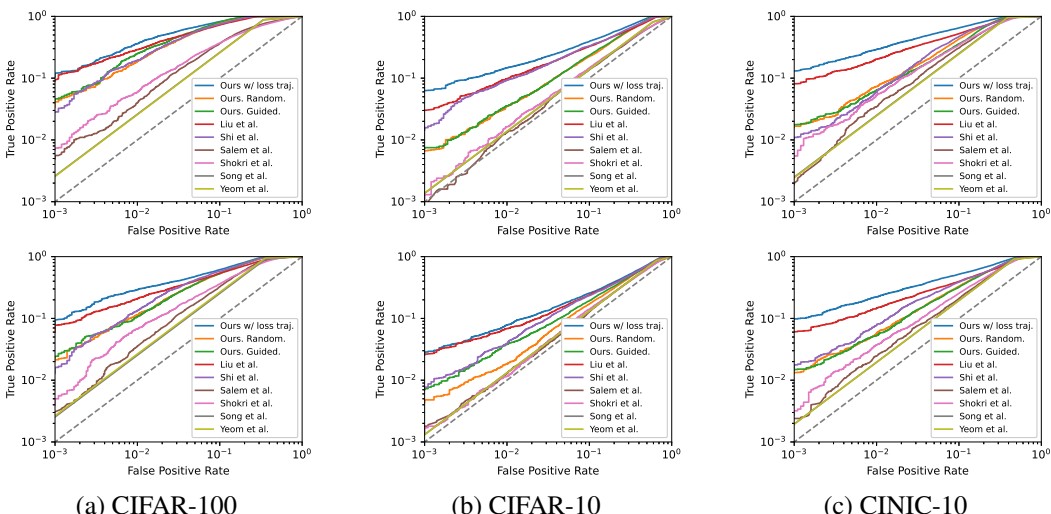

| (a) CIFAR-100 | (b) CIFAR-10 | (c) CINIC-10 |

Figure 3: ROC curves for attacks on three different datasets and two model architectures. The first row shows the results of ResNet-18, and the bottom row shows the results of WideResNet-32.

**Comparison Baselines.** We compare with the state-of-the-art attacks across various adversarial scenarios. In the standard MIA setting as discussed in Section 3.1, we select Yeom et al. (2018), Shokri et al. (2017), Salem et al. (2018), and Song & Mittal (2021) as baselines. For scenarios where the adversary has access to a large supplementary dataset, we choose Liu et al. (2022a) and Shi et al. (2024) as baselines. In scenarios where the adversary can train a large number of shadow models, we adopt Carlini et al. (2022) as our baseline. Additionally, for membership inference attacks on diffusion models, we compare our approach with Duan et al. (2023). Across all these scenarios, our attacks are implemented and evaluated under the same experimental settings as the respective baseline methods. To ensure a fair evaluation, we utilize the same set of shadow and target models throughout our experiments.

**Implementation Details:** To implement our guided-based removal strategy, we first train a classifier model $\mathcal{M}^C$ using the shadow dataset $\mathcal{D}_{shadow}$, which follows the same architecture and training procedures as the shadow model. Next, we train the mask prediction model $\mathcal{P}$ by applying the loss function defined in Eq. 4 on $\mathcal{D}_{shadow}$, with parameters set as $\alpha = 2$ and $\beta = 0.02$. For both the random-based and guided-based removal strategies, we select 50 removal ratios ranging from 0.1 to 1 in equal intervals to generate attack features. In the guided-based strategy, we rank the predicted mask values from lowest to highest, then remove features whose mask value percentiles fall below the specified removal ratio. The attack model is a five-layer MLP with ReLU activation functions, followed by a softmax output layer. We use the Softmax output of MLP as membership scores, indicating the model's prediction confidence. We vary the threshold between 0 and 1 to calculate the TPR at each swept FPR, simulating the adversary's power and error based on these scores.

### 4.2 QUANTITATIVE EVALUATION OF CLASSIFICATION MODELS.

**Evaluation of standard settings.** We have implemented both random-based and guided-based attacks and compared them with baseline methods within the same adversarial setting. The results are presented in Figure 3 and Table 1. We mainly focus the TPR at 0.1% FPR as it is established by Carlini et al. (2022) as the standard in evaluating MIA. We observe that our methods can significantly improve the attack performance at the low FPR, even when employing the random removal-based strategy. This indicates that an adversary can still achieve notable gains with minimal resources by using random removal. Furthermore, our guided removal-based attack outperforms the random strategy, which is attributed to the efficacy of mask prediction models in exploiting the density gaps between members and non-members by accurately estimating the location of important features.

Table 1: Attack performance against different models and datasets. The best and the second-best result at low FPR in each model, dataset and adversarial setting is bold and underlined respectively.

| Target Model | Attack Method | TPR at 0.1% FPR | | | Balanced Accuracy | | | AUC | | |
|---|---|---|---|---|---|---|---|---|---|---|
| | | CIFAR-100 | CIFAR-10 | CINIC-10 | CIFAR-100 | CIFAR-10 | CINIC-10 | CIFAR-100 | CIFAR-10 | CINIC-10 |
| ResNet-18 | Yeom et al. | 0.3% | 0.1% | 0.2% | 0.770 | 0.610 | 0.783 | 0.770 | 0.610 | 0.783 |
| | Shokri et al. | 0.7% | 0.1% | 0.5% | 0.692 | 0.562 | 0.735 | 0.751 | 0.583 | 0.802 |
| | Salem et al. | 0.6% | 0.1% | 0.2% | 0.712 | 0.561 | 0.751 | 0.765 | 0.577 | 0.802 |
| | Song & Mittal | 0.3% | 0.1% | 0.3% | 0.771 | 0.610 | 0.784 | 0.771 | 0.610 | 0.784 |
| | Ours. Random. | 4.1% | 0.7% | 1.7% | 0.810 | 0.639 | 0.760 | 0.938 | 0.732 | 0.853 |
| | Ours. Guided. | **4.6%** | **0.8%** | **1.8%** | 0.825 | 0.646 | 0.780 | 0.946 | 0.737 | 0.840 |
| | Shi et al. | 2.8% | 1.6% | 1.1% | 0.873 | 0.689 | 0.778 | 0.935 | 0.759 | 0.851 |
| | Liu et al. | 8.2% | 2.9% | 7.7% | 0.854 | 0.657 | 0.767 | 0.931 | 0.740 | 0.862 |
| | Ours w/ loss traj. | **12.0%** | **6.3%** | **12.8%** | 0.882 | 0.688 | 0.812 | 0.952 | 0.797 | 0.908 |
| WRN-32 | Yeom et al. | 0.3% | 0.1% | 0.2% | 0.781 | 0.603 | 0.721 | 0.781 | 0.603 | 0.721 |
| | Shokri et al. | 0.5% | 0.2% | 0.3% | 0.732 | 0.560 | 0.684 | 0.799 | 0.585 | 0.739 |
| | Salem et al. | 0.3% | 0.2% | 0.2% | 0.754 | 0.570 | 0.696 | 0.805 | 0.590 | 0.734 |
| | Song & Mittal | 0.3% | 0.1% | 0.2% | 0.780 | 0.603 | 0.722 | 0.780 | 0.603 | 0.722 |
| | Ours. Random. | 2.1% | 0.5% | 1.3% | 0.761 | 0.592 | 0.705 | 0.882 | 0.645 | 0.788 |
| | Ours. Guided. | **2.4%** | **0.7%** | **1.5%** | 0.763 | 0.601 | 0.721 | 0.881 | 0.645 | 0.788 |
| | Shi et al. | 1.6% | 0.8% | 1.8% | 0.810 | 0.614 | 0.726 | 0.886 | 0.663 | 0.797 |
| | Liu et al. | 5.9% | 2.6% | 6.0% | 0.768 | 0.603 | 0.693 | 0.867 | 0.661 | 0.788 |
| | Ours w/ loss traj. | **9.4%** | **2.9%** | **9.8%** | 0.803 | 0.617 | 0.742 | 0.903 | 0.681 | 0.847 |

**Evaluation of settings with large supplementary dataset.** In this setting, we compare our approach with two state-of-the-art baselines: Liu et al. (2022a) and Shi et al. (2024). For Liu et al. (2022a), we utilize the additional datasets to perform model distillation and obtain the loss trajectories. For Shi et al. (2024), we leverage such datasets to train a reference model to calibrate the difficulty of the target data record. Additionally, we implement our attack in this adversarial setting by concatenating our random removal based attack features with the loss trajectories obtained following Liu et al. (2022a) (Ours w/ loss traj.). The results are presented in Figure 3 and Table 1. We observe that in this scenario, our attack performance still surpasses the baselines, especially at the low FPR. This demonstrates that our exploited attack features are applicable to different scenarios.

Table 2: Attack performance in the settings with numerous shadow models.

| Attack Method | Number of Shadow Models | TPR at 0.1% FPR | AUC |
|---|---|---|---|
| LiRA | 8 | 1.8% | 0.591 |
| Ours w/ LiRA | 8 | 3.2% | 0.652 |
| LiRA | 16 | 2.7% | 0.655 |
| Ours w/ LiRA | 16 | 3.8% | 0.676 |
| LiRA | 64 | 7.7% | 0.693 |
| Ours w/ LiRA | 64 | 8.4% | 0.701 |
| LiRA Aug. | 64 | 8.1% | 0.722 |
| Ours w/ LiRA Aug. | 64 | 8.6% | 0.728 |
| R - logit_rescale | 16 | 1.6% | 0.636 |
| R - linear_itp | 16 | 1.2% | 0.641 |
| R - min_linear_logit | 16 | 0.6% | 0.646 |
| R - mean_linear_logit | 16 | 0.5% | 0.649 |
| Ours w/ R | 16 | 2.6% | 0.686 |
| RMIA | 1 | 1.9% | 0.658 |
| Ours w/ RMIA | 1 | 2.5% | 0.679 |
| RMIA | 2 | 3.0% | 0.668 |
| Ours w/ RMIA | 2 | 3.3% | 0.689 |
| RMIA | 4 | 4.1% | 0.673 |
| Ours w/ RMIA | 4 | 4.4% | 0.698 |

**Evaluation of settings with large number of shadow models.** We compare our method with three representative membership inference attacks: LiRA (Carlini et al., 2022), R (Ye et al., 2021), and RMIA (Zarifzadeh et al., 2024). Due to our limited computational resources, we primarily conduct the experiment on CIFAR-10. We reproduced these attacks using their official implementations on WideResNet trained on CIFAR-10 and compared them with our attack in the same settings. To simulate different adversarial capabilities, we train one WideResNet target model and various number of shadow models with random even splits of 50,000 images, each reaching approximately 92% testing accuracy. For Carlini et al. (2022), we focused on the online setting and trained shadow models of 8, 16, and 64. For Ye et al. (2021), we selected their Attack R strategy, which achieves the highest TPR at low FPR. We trained 16 shadow models and implemented their specified attack strategies. For Zarifzadeh et al. (2024), which aims to enhance membership inference performance in low-cost scenarios, we followed their setup by training 1, 2, and 4 shadow models. The results, presented in Table 2, demonstrate that our method outperforms these baseline approaches. Specif-

ically, Compared to SOTA attack RMIA, our method achieves higher TPR and AUC, especially when the number of shadow models. However, training such a large number of shadow models can be highly expensive, especially for large-scale models such as diffusion models.

## 4.3 QUANTITATIVE EVALUATION OF DIFFUSION MODELS

**Experimental setup.** Recently, Duan et al. (2023) proposed SecMI attacks, which achieve state-of-the-art performance on diffusion-based generative models. Their method works by calculating the approximated estimation error of the sample $x$ at a certain step. To adapt our attack to diffusion models, we leverage our proposed random removal strategy to progressively remove features from $x$ with removal ratios from 0.1 to 1, generating a series of perturbed versions $x^*$. We only generate 10 samples, as we found it to be sufficient to significantly improve the attack. We then use these perturbed samples to conduct the membership inference.

To ensure a fair comparison, we follow the same experimental settings as their method. We conduct the experiment on the DDPM Ho et al. (2020) model, using the CIFAR-100 and CIFAR-10 datasets. They propose two attack methods: the statistic-based inference (SecMI-Stat) and the neural network-based inference (SecMI-NNs), with the neural network-based strategy performing better. This strategy takes the pixel-wise absolute value of the estimation error and trains an NN to predict the membership status. We

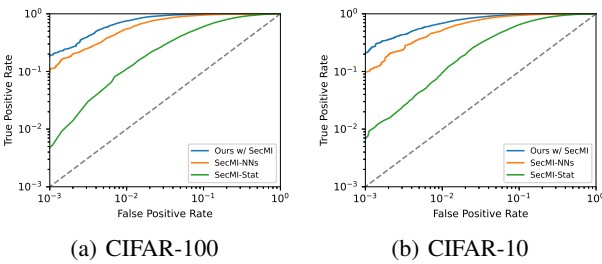

(a) CIFAR-100        (b) CIFAR-10

Figure 4: ROC curves of different attacks against DDPM trained on CIFAR-100 and CIFAR-10.

calculate the estimation error of every $x^*$ and concatenate them as input features. For evaluation, their method uses the attack success rate (ASR) instead of balanced accuracy, which we also follow.

**Evaluation results.** The results are shown in Figure 4 and Table 3. Our method significantly improves the attack performance at the low FPR. For example, our method achieves a TPR of 18.4% at 0.1% FPR, compared to 9.7% in SecMI-NNs and 0.5% in SecMI-Stat. Our method also achieves better AUC and ASR. This demonstrates the applicability of our approach to diffusion models.

Table 3: Comparison of attack performance against DDPM trained on CIFAR-100 and CIFAR-10.

| Dataset | Method | TPR at 0.1% FPR | ASR | AUC |
|---|---|---|---|---|
| | SecMI-Stat | 0.5% | 0.801 | 0.873 |
| CIFAR-100 | SecMI-NNs | 9.7% | 0.930 | 0.975 |
| | Ours w/ SecMI | 18.4% | 0.960 | 0.989 |
| | SecMI-Stat | 0.6% | 0.816 | 0.887 |
| CIFAR-10 | SecMI-NNs | 9.6% | 0.922 | 0.973 |
| | Ours w/ SecMI | 20.4% | 0.940 | 0.983 |

## 4.4 ABLATION STUDY

In this section, we evaluate the impact of various removal operations and steps, followed by an assessments in practical adversarial settings. Further ablation studies are provided in Appendix C.

**Different Removal Operations:** In this section, we explore various removal operations commonly used in the explainable machine learning (Rong et al., 2022; Yoon et al., 2018; Fong & Vedaldi, 2017): channel mean, Gaussian Noise, Gaussian Blur, and Generative Adversarial Network (GAN). We compare these methods against our Noisy Linear Imputation strategy in terms of attack performance. Specifically, the channel mean method fills the removed areas using the average color intensity of each channel. Gaussian noise is introduced with a mean of 0 and a standard deviation of 0.1. Gaussian blur is applied using a kernel size of 11 and a standard deviation of 5. For the GAN approach, we follow the approach described in Kachuee et al. (2020) to train the model. The experiments were conducted on a ResNet-18 model trained on the CIFAR-100 dataset, and the results

are presented in Table 4. The findings indicate that Noisy Linear Imputation outperforms the other methods, which is attributed to its ability to mitigate the distribution shift and adversarial artifact.

Table 4: Attack performance of various removal operations.

| Removal Operation | TPR at 0.1% FPR | Balanced Accuracy | AUC |
|---|---|---|---|
| Channel Mean | 2.7% | 0.703 | 0.922 |
| Gaussian Noise | 3.5% | 0.730 | 0.929 |
| Gaussian Blur | 3.4% | 0.771 | 0.930 |
| GAN | 3.9% | 0.809 | 0.936 |
| Noisy Linear Imputation | 4.1% | 0.810 | 0.938 |

**Number of Removal Steps.** We conducted removal operations across 50 steps with ratios ranging from 0.1 to 1 during our main evaluation. In this section, we explore the impact of the number of steps on the attack performance, as it directly affects the number of queries made to the model. We conducted the experiment on WideResNet-32 trained on the CIFAR-100 dataset. The experimental results are presented in Table 5. We observe that the overall performance improves with an increasing number of steps, as it provides more fine-grained information about the changes in the model's output. Nevertheless, our method also demonstrates decent performance at lower numbers of steps. This demonstrates that our method is query-efficient, requiring only an additional few dozen queries.

Table 5: Attack performance of different removal steps.

| Removal Step Number | TPR at 0.1% FPR | Balanced Accuracy | AUC |
|---|---|---|---|
| 5 | 1.9% | 0.750 | 0.877 |
| 10 | 2.2% | 0.737 | 0.880 |
| 20 | 2.4% | 0.748 | 0.880 |
| 50 | 2.4% | 0.763 | 0.881 |

**Disjoint Distribution between Shadow and Target Dataset.** In previous experiments, we assumed that the adversary possesses shadow datasets that share the same distribution as the target dataset, which may not hold in practical settings. In this section, we consider a scenario where the adversary has access to a dataset that is disjoint from the target datasets. Specifically, we utilize the CINIC-10 dataset, which merges CIFAR-10 with an additional 210k images from ImageNet that match classes contained in CIFAR-10. For this experiment, the target model was trained using the CIFAR-10 portion, while the shadow model was trained on the ImageNet portion, representing a different distribution (*i.e.*, diff. dist.). We also implement baseline methods in this scenario for comparison. Additionally, we implement an attack where both models were trained on the CIFAR-10 dataset (*i.e.*, same dist.). The ResNet-18 architecture was used for this experiment, with results presented in Figure 5, which indicate that our attack's performance remains robust when using a disjoint dataset and notably outperforms the baseline methods in this setting.

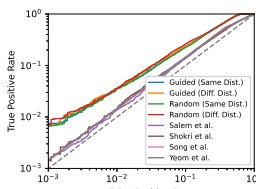

Figure 5: ROC curves on scenarios of disjoint shadow and target distribution.

## 5 CONCLUSION

In this paper, we introduce a novel membership inference attack that adapts effectively to various models, datasets, and adversarial settings, achieving significant improvements in attack performance across different scenarios with minimal resource requirements. The key insight behind our approach is that member samples tend to reside in high-density regions of the learned feature space. Consequently, even with partial feature removal, the model is likely to maintain higher confidence in these samples. Building on this, we propose a progressive feature removal technique, where features are incrementally removed from the input. We then leverage the model outputs within a progressively increasing removal ratio as attack features to conduct membership inference. Extensive experiments across multiple datasets, models, and threat models demonstrate that our approach consistently surpasses state-of-the-art methods in a variety of scenarios.

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

## A    EXPERIMENTAL SETTINGS

### A.1    DATA SPLITS ON DIFFERENT DATASETS

We follow the dataset splitting method in (Liu et al., 2022a; 2024). The detailed splits for different datasets are provided in Table 6. The target model is trained using $\mathcal{D}_{target}^{train}$, and the samples from this set serve as the member samples of $\mathcal{M}$, while the samples from $\mathcal{D}_{target}^{test}$ are used as the non-member samples. Similarly, the shadow model is trained using $\mathcal{D}_{shadow}^{train}$, and the samples from this set serve as the member samples of the model $\mathcal{M}^S$, while the samples from $\mathcal{D}_{shadow}^{test}$ are used as the non-member samples. The auxiliary dataset $\mathcal{D}_{aux}$ is utilized for model distillation for Liu et al. (2022a) and for training the reference model for Shi et al. (2024).

### A.2    TRAINING CONFIGURATIONS

In the standard adversarial setting, we use one shadow model and one target model. For settings with large supplementary datasets, we follow the approach of Liu et al. (2022), applying knowledge distillation for both shadow and target models and storing intermediate model checkpoints. For settings with a large number of shadow models, we follow the strategy of Carlini et al. (2022) and employ 8, 16, and 64 shadow models alongside a single target model. We train each model for 100 epochs using an initial learning rate of 0.1. Additionally, we leverage a cosine annealing schedule to gradually reduce the learning rate. To further improve model generalization, we employ standard data augmentations and apply a weight decay rate of 0.0001. The training and testing accuracies for

various models across different datasets are presented in Table 7. We have repeated the experiments 10 times with different random samplings of the datasets, except for the experiments based on Carlini et al. (2022) due to the large computational costs associated with training shadow models.

Table 6: Data splits on different datasets.

| Dataset | $\mathcal{D}_{target}^{train}$ | $\mathcal{D}_{target}^{test}$ | $\mathcal{D}_{shadow}^{train}$ | $\mathcal{D}_{shadow}^{test}$ | $\mathcal{D}_{aux}$ |
|---|---|---|---|---|---|
| CIFAR-100 | 10000 | 10000 | 10000 | 10000 | 20000 |
| CIFAR-10 | 10000 | 10000 | 10000 | 10000 | 20000 |
| CINIC-10 | 10000 | 10000 | 10000 | 10000 | 220000 |

Table 7: Training and testing accuracy for various model architectures on different datasets.

| Model | Dataset | Training Accuracy | Testing Accuracy |
|---|---|---|---|
| | CIFAR-100 | 1.000 | 0.443 |
| ResNet-18 | CIFAR-10 | 0.996 | 0.763 |
| | CINIC-10 | 0.999 | 0.625 |
| | CIFAR-100 | 0.999 | 0.584 |
| WideResNet-32 | CIFAR-10 | 0.992 | 0.824 |
| | CINIC-10 | 0.998 | 0.704 |

# B    MASK PREDICTION MODEL

The rationale of using a mask prediction model instead of directly optimize the mask for every input stems from several considerations. First, under the black-box threat model assumed in this work, the adversary does not have access to gradient information from the target model. As a result, directly optimizing the mask $m$ using the proposed loss function is not feasible because gradient-based updates cannot be performed. Additionally, while it is possible to use a shadow model to estimate the mask, optimizing $m$ for each sample individually would be computationally expensive. This process involves multiple components of the proposed loss function and requires numerous backpropagation steps for every sample. By contrast, training a U-Net model allows for efficient mask generation. After training, the U-Net produces the mask in a single forward pass, significantly improving the computational efficiency of the membership inference attack. Also, the U-Net model is particularly well-suited for this task due to its established effectiveness in image generation, *e.g.*, in DDPM (Ho et al., 2020). U-Net utilizes feature maps at multiple resolutions to generate sharp and precise outputs. Additionally, since the threat models assume that the adversary is aware of the target model's architecture and has access to an auxiliary dataset that shares the same distribution as the target model's training data. Under this assumption, the shadow model, trained on the auxiliary dataset, is designed to closely replicate the behavior of the target model. The mask prediction model, trained using the outputs of the shadow model for both members and non-members, leverages this similarity. As the shadow model and target model exhibit comparable decision-making patterns, the model is expected to generalize effectively to predict masks for samples from the target model.

The proposed mask prediction model adopts an encoder-decoder U-Net architecture (Ronneberger et al., 2015). The encoder is based on a pre-trained ResNet-50, which consists of an initial convolutional layer followed by four scale levels, each containing multiple residual blocks. The encoder progressively downsamples and extracts features from the input image. To incorporate class-specific information into the model, the class label of the input image is embedded into a vector. This class embedding is then element-wise multiplied with the features from the last encoder scale. By doing so, the model can learn to generate masks that are conditioned on the specific class of the input image. The decoder aims to recover spatial details by gradually upsampling the encoded features. It comprises three upsampling blocks, each corresponding to a scale level in the encoder. The upsampling blocks receive the output from the previous block and integrate corresponding encoder features via skip connections. Finally, the output of the decoder's last layer is passed through a convolutional layer with two output channels. The absolute values of these channels are normalized to produce the final mask prediction.

# C ADDITIONAL ABLATION STUDY

**Different Removal Strategy.** As discussed in Section 3.4, we employ random grouping to tackle the challenges of high dimensionality. In this section, we empirically assess the impact of other strategies on our attack performance. We have chosen five widely used superpixel segmentation methods for comparison: Felzenszwalb (Felzenszwalb & Huttenlocher, 2004), Quickshift (Vedaldi & Soatto, 2008), SLIC (Achanta et al., 2012), Watershed (Neubert & Protzel, 2014), and DISF (Belém et al., 2020). Additionally, we consider saliency map methods. Since we assume a black-box threat model for the target model, we cannot utilize methods that require white-box knowledge, such as GradCAM (Selvaraju et al., 2017). Therefore, we selected two widely used saliency methods that work in black-box settings: LIME (Ribeiro et al., 2016) and SHAP (Lundberg & Lee, 2017). We compare these methods with our random-based removal and guided-removal strategies. We conducted the experiment on ResNet-18 trained on CIFAR-100. The results are presented in Table 8. The findings demonstrate that all of these methods exhibit effectiveness compared to the baselines in Table 1, with our guided removal strategy achieving the best performance. While the saliency map methods only perform slightly lower than our approach, we want to highlight that these black-box saliency methods are generally very computationally expensive. They require a large number of queries to approximate feature importances—typically several hundred to a thousand queries per sample to derive a saliency map. In contrast, our method is very efficient: training a mask prediction model takes less than 2 minutes on an Nvidia 4090 GPU, and it does not require access to the target model to derive the mask.

Table 8: Attack performance of different removal strategies.

| Removal Strategy | TPR at 0.1% FPR | Balanced Accuracy | AUC |
|---|---|---|---|
| Felzenszwalb (Felzenszwalb & Huttenlocher, 2004) | 3.7% | 0.760 | 0.929 |
| Quickshift (Vedaldi & Soatto, 2008) | 3.0% | 0.717 | 0.926 |
| SLIC (Achanta et al., 2012) | 3.1% | 0.788 | 0.930 |
| Watershed (Neubert & Protzel, 2014) | 3.5% | 0.804 | 0.931 |
| DISF (Belém et al., 2020) | 3.0% | 0.772 | 0.935 |
| LIME (Ribeiro et al., 2016) | 4.2% | 0.821 | 0.940 |
| SHAP (Lundberg & Lee, 2017) | 4.0% | 0.813 | 0.938 |
| Ours. Random. | 4.1% | 0.810 | 0.938 |
| Ours. Guided. | 4.6% | 0.825 | 0.946 |

**Evaluation on Differential Privacy.** Differential privacy (DP) (Abadi et al., 2016) is a widely used mechanism to defend against membership inference attacks by providing a rigorous bound on the ability to distinguish between two neighboring datasets that differ by only one data sample. We utilize the Fast Differential Privacy library (Bu et al., 2023) to implement DP, which achieves notable privacy-preserving performance while minimizing computational cost. We set the per-sample gradient clipping threshold to automatic and choose MixOpt as our gradient clipping mode, applying the clipping style to all layers. The experiments are conducted on ResNet-18 trained on the CIFAR-100 dataset, with varying privacy budget values ($\epsilon$) of 1, 100, and 1000. The results are detailed in Table 9. We observe that stricter privacy budgets significantly reduce both the TPR and Balanced Accuracy of the attack, indicating effective privacy protection. However, implementing differential privacy impacts the model's accuracy; for instance, there is a notable drop in top-1 accuracy of 0.28 when $\epsilon = 1000$. These results highlight the challenging trade-off between defensive effectiveness and performance utility in the application of differential privacy.

Table 9: Model accuracy and attack performance under DP with different privacy budget $\epsilon$.

| Privacy Budget | Top-1 Acc. Drop | Top-5 Acc. Drop | TPR at 0.1% FPR | Balanced Acc. |
|---|---|---|---|---|
| 1 | 0.408 | 0.599 | 0.0% | 0.499 |
| 100 | 0.331 | 0.368 | 0.1% | 0.508 |
| 1000 | 0.280 | 0.268 | 0.2% | 0.514 |

**Different Architectures between Shadow and Target Model.** We now relax another assumption in the threat model, where the adversary knows the target model architecture. To this end, we use

ResNet-18 as the target model architecture and vary the architectures of the shadow models using VGG-16 (Simonyan & Zisserman, 2014), DenseNet-161 (Huang et al., 2017), WideResNet-32, and ResNet-18. We conduct the experiment using CIFAR-100, the results are given in Figure 6. We observe that when the shadow model shares the same architecture as the target model, our method achieves the best performance. Additionally, the attack performance of other networks also achieves decent results. This demonstrates that our attack remains effective even when the adversary lacks knowledge of the target model's architecture.

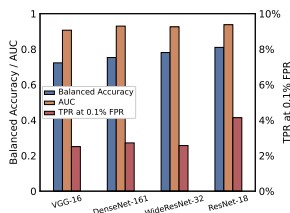

Figure 6: The impact of different model architectures on the attack performance.

**Attack on Other Data Modalities.** We primarily focus on the image modality to demonstrate the effectiveness of our attack in this paper, as it is the most studied modality in existing works (Liu et al., 2022a; Shokri et al., 2017; Song & Mittal, 2021). In this section, we extend our attacks to other modalities. Our attacks are based on the insight that members exhibit more redundancy in the features learned by ML models, as discussed in Section 3.2. While different feature removal strategies are needed to tailor this attack to other data types, this insight generally enhances attack performance across various modalities. We conducted a preliminary experiment using the tabular dataset Adult (Becker & Kohavi, 1996) to test this hypothesis. We employed a three-layer MLP model as the target model, consisting of one hidden layer with a ReLU activation function, followed by a Softmax layer. We iteratively removed features by setting them to zero and recorded the corresponding confidence score drop as attack features. The results, presented in Table 10, demonstrate a performance boost compared to baselines. Our attack achieved a 0.8% TPR at a 0.1% FPR, compared to a 0.2% TPR by Shokri et al. (2017) and a 0.1% TPR by other methods at a 0.1% FPR.

Table 10: Attack performance on other data modalities.

| Method | TPR at 0.1% FPR | Balanced Accuracy | AUC |
|---|---|---|---|
| Yeom et al. | 0.1% | 0.553 | 0.545 |
| Song et al. | 0.1% | 0.554 | 0.546 |
| Salem et al. | 0.1% | 0.518 | 0.514 |
| Shokri et al. | 0.2% | 0.523 | 0.519 |
| Our Method | 0.8% | 0.563 | 0.594 |

## D EMPIRICAL EVALUATION OF FEATURE DENSITY

To empirically validate the feature density gaps between members and non-members, we conducted experiments by extracting features from the layer immediately before the final classification layer for both member and non-member samples. We calculated the average L2 distance of each sample to its top 5 nearest neighbors in the feature space. The experiments were repeated across different models and datasets, and the mean L2 distance is as follows: (1) ResNet-18 trained on CIFAR-100: Members: $71.11 \pm 1.08$, non-members: $78.86 \pm 0.97$; (2) ResNet-18 trained on CIFAR-10: Members: $13.42 \pm 0.22$, non-members: $15.52 \pm 0.34$; (3) WideResNet trained on CIFAR-100: Members: $54.34 \pm 0.78$, non-members: $60.65 \pm 0.66$; (4) WideResNet trained on CIFAR-10: Members: $2.99 \pm 0.04$, non-members: $3.22 \pm 0.03$. These results consistently indicate that member samples have lower mean L2 distances to their nearest neighbors compared to non-member samples, reflecting a higher feature density.

## E ATTACK COST

In real-world scenarios, the cost of executing an attack is a crucial factor that needs to be considered alongside its effectiveness. In certain use cases, a cost-efficient strategy, even if it sacrifices some performance, can be more practical and advantageous (Shi et al., 2024). We address resource inefficiency in existing methods from two main perspectives: (1) Many existing methods (*e.g.*, Liu et al. (2022a), Shi et al. (2024)) require large auxiliary datasets with the same distribution of the target

model's training data (in addition to shadow datasets), which is often impractical in real-world scenarios. (2) Some approaches, such as Carlini et al. (2022), require training a large number of shadow models, leading to significant computational costs. We provide a comparison of the additional number of trained models and total training time in Table 13, demonstrating that our method achieves better computational efficiency. Additionally, by comparing Ours. Random. and Ours. Guided., we show that the mask prediction model incurs minimal additional training time.

Table 11: Computational cost of different methods.

| Method | Number of Trained Models | Training Time |
|---|---|---|
| Shi et al. | 2 | 47.2 min |
| Liu et al. | 2 | 51.7 min |
| Carlini et al. | 64 | 1499.7 min |
| Ours. Random. | 1 | 23.4 min |
| Ours. Guided. | 2 | 24.6 min |

## F  ATTACK PERFORMANCE WITHOUT SHADOW MODELS

A significant strength of our attack is its ability to be executed on pre-trained models without requiring the training of shadow models. To validate this point, we conducted preliminary experiments following the same settings as the population attacks in Ye et al. (2021), without training shadow models. We implemented both attacks on ResNet-18 trained with CIFAR-100. The population attack achieved a TPR of only 0.4% at a 0.1% FPR with an AUC of 0.771. In contrast, our method achieved a TPR of 1.9% and an AUC of 0.916 under the same settings, significantly improving upon the population attack. This demonstrates the advantage of our approach, which does not require shadow models and is particularly beneficial for large-scale models, thereby significantly reducing computational costs.

## G  ATTACK PERFORMANCE WITH PRE-TRAINED MODELS

Pre-training followed by fine-tuning has become a widely adopted paradigm in modern machine learning (Liu et al., 2022b). To evaluate the performance of our attack under this setting, we conducted an experiment using a pre-trained DenseNet-161 model on ImageNet. We applied transfer learning to adapt this model on CIFAR-100 and then evaluated our random-based attack against the baseline methods in standard adversarial settings. The results, presented in Table 12, clearly show that our attack remains effective even when the defender employs a pre-trained feature extractor during target model training.

Table 12: Comparison of attack performance with a pre-trained models.

| Attack Method | TPR at 0.1% FPR | Balanced Accuracy | AUC |
|---|---|---|---|
| Yeom et al. | 0.2% | 0.676 | 0.676 |
| Shokri et al. | 0.5% | 0.576 | 0.599 |
| Salem et al. | 0.3% | 0.571 | 0.596 |
| Song & Mittal | 0.2% | 0.677 | 0.677 |
| Ours. Random. | 4.7% | 0.866 | 0.941 |

## H  ANALYSIS OF FEATURE DIFFERENCE OF MEMBER AND NON-MEMBERS

To explore the differences in learned features between member and non-member samples, we conducted an experiment using a common set of samples from CIFAR-100. In the first scenario, the samples were used to train a model, thereby serving as member samples. In the second scenario, the same samples were excluded from the training process, making them non-member samples. We generated saliency maps for these samples in both scenarios using GradCAM and measured their similarity using the Structural Similarity Index Measure (SSIM) (Wang et al., 2004). The SSIM value was computed to be $0.51 \pm 0.17$. Figure 8 presents several demonstration examples, revealing

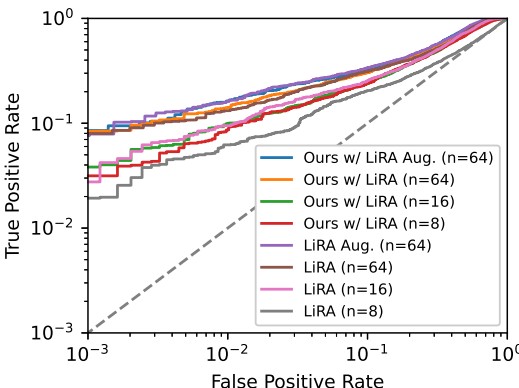

Figure 7: ROC curves for LiRA attack and our attack under the different number of shadow models.

Table 13: The mean and standard deviation of different performance metrics.

| Attack Method | TPR at 0.1% FPR | Balanced Accuracy | AUCSection |
|---|---|---|---|
| Yeom et al. | 0.3±0.0% | 0.770±0.022 | 0.770±0.002 |
| Shokri et al. | 0.7±0.1% | 0.692±0.015 | 0.751±0.004 |
| Salem et al. | 0.6±0.1% | 0.712±0.024 | 0.765±0.003 |
| Song & Mittal | 0.3±0.0% | 0.771±0.026 | 0.771±0.002 |
| Ours. Random. | 4.1±0.2% | 0.810±0.030 | 0.938±0.006 |
| Ours. Guided. | 4.6±0.3% | 0.825±0.026 | 0.946±0.003 |
| Shi et al. | 2.8±0.2% | 0.873±0.020 | 0.935±0.003 |
| Liu et al. | 8.2±0.5% | 0.854±0.014 | 0.931±0.007 |
| Ours w/ loss traj. | 12.0±0.5% | 0.882±0.016 | 0.952±0.003 |

clear disparities between the saliency maps of member and non-member samples. Specifically, the saliency maps for member samples focus more prominently on key semantic features (*i.e.*, the main object critical for classification), whereas non-member saliency maps either fail to concentrate on or only partially engage with these critical features.

These observations explain the efficacy of our guided-based approach. By estimating feature importance and removing the least important features, our method minimizes the likelihood of affecting critical features in member samples. In contrast, non-member samples, which exhibit a weaker focus on key features, are less robust to feature removal.

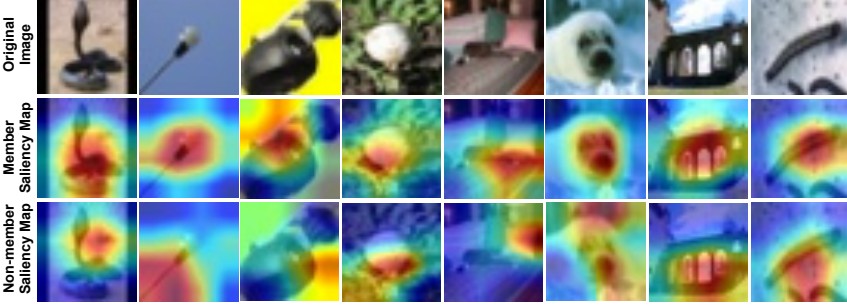

Figure 8: The comparison of the saliency map between member and non-member samples. The first row shows the original images, the second row shows the saliency maps when they are member samples, and the third row shows the saliency maps when they are non-member samples.

