# OpenReview forum: "Less is More: Exploiting Feature Density for Enhanced Membership Inference Attacks"
_ICLR.cc/2025/Conference — Submitted to ICLR 2025_

### Official Review · Reviewer_dYkK · 2024-10-27

**Soundness:** 3
**Presentation:** 2
**Contribution:** 2
**Rating:** 6
**Confidence:** 4

**Summary:**

This paper proposes a new Membership Inference Attack that leverages feature density gaps between member and non-member data points, allowing the attack to significantly improve true positive rates at low false positive rates. This method requires far fewer resources compared to traditional approaches, as it does not depend on large auxiliary datasets or numerous shadow models. By systematically removing features and analyzing changes in model confidence, the new approach can effectively discriminate between members and non-members, demonstrating superior performance across various datasets and model architectures in the process.

**Strengths:**

1. Membership inference attack is a popular and interesting topic.
2. The proposed method is interesting and the design intuition makes sense to me.
3. Experiments are detailed and sufficient。

**Weaknesses:**

1. Lack of important baselines such as [1] and [2]
2. Some parts of figures such as figure 2 and 4 and the experiments are unclear
3. This paper's contribution is kind of overclaimed.




[1] Ye, Jiayuan, et al. "Enhanced membership inference attacks against machine learning models." Proceedings of the 2022 ACM SIGSAC Conference on Computer and Communications Security. 2022.
[2]  Zarifzadeh, Sajjad, Philippe Liu, and Reza Shokri. "Low-Cost High-Power Membership Inference Attacks." Forty-first International Conference on Machine Learning. 2024.

**Questions:**

I think the idea is good but I still have some questions:

1. I think the proposed method should be compared to well-known attacks such as [1] and current SOTA such as [2] to make the claim this work is SOTA. What will be the results when compared to [1] and [2] proposed methods? Especially, [2] also claims that their proposed method can be low cost making it a good baseline to this work.

2. I am curious that why the balanced accuracy is approximately 0.7, while the AUC reaches around 0.9. Although theoretically possible, I haven't encountered such a significant discrepancy in practical applications, where attack accuracy generally aligns closely with attack AUC. Could the author clarify this anomaly or consider releasing the source code to provide further insights?

3. This paper claims that the proposed method is resource-efficient but they are not very obvious in the paper presentation. Using around 10k for training and testing and rest for MIA is a common thing in the literature. I do not see why this experiment setting is resource-efficient. For the number of shadow models, it may be required a lot for LiRA attack, but it is not that big for other attacks. Especially, for population attacks in [1], you do not need to train the shadow model.  BTW, figure 4 is too small to distinguish the difference.

4. I also wonder if the defender uses a pre-trained feature extractor during target model training, will the proposed attack still work? Or in other words, is your proposed method limited to a ResNet-like model and training from scratch setting?

I will consider changing the score based on the response.

---

> ### Author Response · Authors · 2024-11-20
> **Response to Reviewer dYkK (1/2)**
>
> Thank you for your constructive feedback. We provide detailed responses to your comments.
>
> **Q1: I think the proposed method should be compared to well-known attacks such as [1] and current SOTA such as [2] to make the claim this work is SOTA. What will be the results when compared to [1] and [2] proposed methods? Especially, [2] also claims that their proposed method can be low cost making it a good baseline to this work.**
>
> Thank you for your valuable feedback. We have conducted additional experiments to compare our method with the newer, stronger attacks mentioned in [1] and [2]. Specifically, we reproduced these attacks using their official implementations on WideResNet trained on CIFAR-10 and compared them with our attack in the same settings. For the attack in [1], we selected Attack R, which achieves the highest TPR at low FPR. We trained 16 shadow models, and the experimental results with different their attack strategies are as follows:
>
> | Attack Method | TPR at 0.1% FPR | AUC |
> |:----------:|:----------:|:----------:|
> | logit_rescale  | 1.6%  | 0.636  |
> | linear_itp  | 1.2%   | 0.641   |
> | min_linear_logit  | 0.6%   | 0.646   |
> | mean_linear_logit  | 0.5%   | 0.649   |
> | Ours. Random.  | 2.6%   | 0.686   |
>
>
> For RMIA [2], which focuses on improving membership inference performance in low-cost settings, we followed their experimental setup by training 1, 2, and 4 shadow models. The results are as follows:
>
> | Method | Number of Shadow Models | TPR at 0.1% FPR | AUC   |
> |---------|:----------:|:----------:|:----------:|
> |         RMIA        | 1                        | 1.9%             | 0.658 |
> |        RMIA         | 2                        | 3.0%             | 0.668 |
> |         RMIA        | 4                        | 4.1%             | 0.673 |
> |        Ours. Random.         | 1                        | 2.5%             | 0.679 |
> |        Ours. Random.         | 2                        | 3.3%             | 0.689 |
> |        Ours. Random.         | 4                        | 4.4%             | 0.698 |
>
>
> As the tables indicate, our method outperforms both attacks from [1] and [2]. For the attack in [1], we improved the TPR at 0.1% FPR from 1.6% to 2.6%. Compared to SOTA attack RMIA [2], our method achieves higher TPR and AUC, especially when the number of shadow models is low. We have revised Section 4.2 to include these results.
>
>
> **Q2: I am curious that why the balanced accuracy is approximately 0.7, while the AUC reaches around 0.9. Although theoretically possible, I haven't encountered such a significant discrepancy in practical applications, where attack accuracy generally aligns closely with attack AUC. Could the author clarify this anomaly or consider releasing the source code to provide further insights?**
>
> Thanks for pointing this out. To maintain consistency with previous evaluations (Shokri et al., Salem et al., Liu et al.), we calculated the balanced accuracy using a fixed threshold of 0.5 on the softmax output of the attack model to determine membership status. In contrast, the ROC curve and the corresponding AUC are obtained by varying this threshold across all possible values between 0 and 1, which is aligned with [1] [2]. This process allows us to compute the TPR and FPR at different thresholds, capturing the model's performance across all operating points. We have revised Section 4.1 to make it clearer.
>
> Our attack model achieves a high TPR at low FPR values, resulting in a steep initial slope in the ROC curve. This steep rise means the ROC curve quickly diverges from the diagonal line representing random guessing and moves toward the top-right corner. This characteristic significantly contributes to a higher AUC. When calculating balanced accuracy at the fixed threshold of 0.5, we may not be operating at the model's optimal point on the ROC curve. If the optimal threshold that maximizes balanced accuracy differs from 0.5, using this fixed threshold can lead to a lower observed balanced accuracy. This discrepancy is also observed in existing studies, such as Carlini et al. (2022), where an AUC of 0.925 is reported, yet the accuracy is only 0.826. We will publicly release our code to facilitate reproducibility.

---

> > ### Comment · Reviewer_NyRP · 2024-12-01
> >
> > Thank you for reading my review and updating the paper. I have updated my score.

---

> > > ### Author Response · Authors · 2024-12-01
> > > **Response to Reviewer NyRP**
> > >
> > > Thank you for your feedback and for updating the score! I appreciate your insights and am glad the revisions addressed your concerns.

---

> ### Author Response · Authors · 2024-11-20
> **Response to Reviewer dYkK (2/2)**
>
> **Q3: This paper claims that the proposed method is resource-efficient but they are not very obvious in the paper presentation. Using around 10k for training and testing and rest for MIA is a common thing in the literature. I do not see why this experiment setting is resource-efficient. For the number of shadow models, it may be required a lot for LiRA attack, but it is not that big for other attacks. Especially, for population attacks in [1], you do not need to train the shadow model. BTW, figure 4 is too small to distinguish the difference.**
>
> Thank you for highlighting this point. To clarify this, we provide explicit comparisons to baseline methods in terms of resource efficiency. The table below compares the computational resource requirements (number of trained models and training time) for our method and state-of-the-art methods, which demonstrates that our method could achieve better computational efficiency compared to previous works. Although population attacks do not require shadow model training, they necessitate a large number of queries to the target model to determine the threshold, which also incurs high resource costs. Moreover, our attack substantially outperforms population attacks. We implemented a population attack under the same settings on ResNet-18 trained with CIFAR-100, achieving a TPR of only 0.4% at 0.1% FPR with an AUC of 0.771, compared to our method’s TPR of 4.6% and AUC of 0.946.
>
> | Attack Method | Number of Trained Models | Training Time |
> |:----------:|:----------:|:----------:|
> | Shi et al.  | 2  | 47.2 min  |
> | Liu et al.  | 2   | 51.7 min   |
> | Carlini et al.  | 64   | 1499.7 min   |
> | Ours. Random.  | 1   | 23.4 min   |
> | Ours. Guided.  | 2   | 24.6 min   |
>
> Furthermore, beyond computational resources, our method is also resource-efficient in terms of auxiliary data requirements. Many existing methods (e.g., Liu et al., 2022; Shi et al., 2024) require large auxiliary datasets with the same distribution as the target model's training data, in addition to shadow datasets. This requirement is often impractical in real-world scenarios. In contrast, our method does not rely on such large auxiliary datasets. We have revised Appendix E to include a comparison of attack costs.
>
> Additionally, we acknowledge your feedback regarding Figure 4. We have enlarged the figure and included the following table to better clarify our results. We have revised Section 4.2 and Figure 7 to make it clearer.
>
> |           Method           | TPR at 0.1% FPR |   AUC   |
> |:--------------------------:|:----------------:|:-------:|
> |       LiRA (n=8)          |       1.8%       |  0.591  |
> |    Ours w/ LiRA (n=8)     |       3.2%       |  0.652  |
> |       LiRA (n=16)         |       2.7%       |  0.655  |
> |    Ours w/ LiRA (n=16)    |       3.8%       |  0.676  |
> |       LiRA (n=64)         |       7.7%       |  0.693  |
> |    Ours w/ LiRA (n=64)    |       8.4%       |  0.701  |
> |   LiRA Aug. (n=64)        |       8.1%       |  0.722  |
> | Ours w/ LiRA Aug. (n=64)  |       8.6%       |  0.728  |
>
>
> **Q4: I also wonder if the defender uses a pre-trained feature extractor during target model training, will the proposed attack still work? Or in other words, is your proposed method limited to a ResNet-like model and training from scratch setting?**
>
> Our method is not limited to ResNet-like models. In fact, we have demonstrated the effectiveness of our attacks even on diffusion models, which are generative and structurally distinct from ResNet-based classifiers. To address the reviewer’s question about the use of a pre-trained feature extractor, we conducted an additional experiment using a pre-trained DenseNet-161 model on ImageNet. We applied transfer learning to adapt this model on CIFAR-100 and then evaluated our random-based attack against the baseline methods in standard adversarial settings. The results, presented in the following tables, clearly show that our attack remains effective even when the defender employs a pre-trained feature extractor during target model training. We have revised Appendix G to include these results.
>
> |     Attack Methods     | TPR at 0.1% FPR | Balanced Accuracy |   AUC   |
> |:----------------------:|:----------------:|:-----------------:|:-------:|
> |      Yeom et al.       |      0.2%       |       0.676       |  0.676  |
> |      Shokri et al.     |      0.5%       |       0.576       |  0.599  |
> |      Salem et al.      |      0.3%       |       0.571       |  0.596  |
> |     Song & Mittal      |      0.2%       |       0.677       |  0.677  |
> |     Ours. Random.      |      4.7%       |       0.866       |  0.941  |

---

> ### Author Response · Authors · 2024-11-25
>
> Dear Reviewer dYkK,
>
> Thank you once again for your thoughtful feedback. We have provided a detailed response to your questions and hope it sufficiently addresses your concerns. With the interactive discussion period closing soon, please don't hesitate to reach out if you need any additional information or clarification.

---

### Official Review · Reviewer_NyRP · 2024-10-29

**Soundness:** 2
**Presentation:** 2
**Contribution:** 2
**Rating:** 6
**Confidence:** 5

**Summary:**

Membership Inference Attacks (MIAs) aim to determine whether a model was trained on a specific data point. Most MIAs perform poorly at low false positive rates (FPR) because non-member samples often resemble member samples in terms of model output. This paper introduces a novel MIA that leverages feature density to ascertain membership information. The attack demonstrates that membership inference can be determined by progressively removing features from a sample: the confidence scores of non-members decrease faster than those of members as features are removed. The authors evaluate their attack on both image classification and diffusion models.

**Strengths:**

There is a lot I appreciated about this work.

**The attack mechanism is intuitive and well-explained:**
I really like the intuition behind this attack and how manipulating features can reveal membership information. It’s a refreshing approach, distinct from existing methods.

**Does not require training shadow models:**
Unlike most existing MIAs that require training a shadow model, this attack does not, as it operates within the feature space. Consequently, this attack (or a variation of it) could be executed on a large pre-trained model. While this is a major strength, it is not emphasized by the authors.

**Evaluation across multiple datasets (CIFAR10/100 and CINIC):**
The authors perform experiments on standard datasets commonly used in MIA literature and test the attack across multiple architectures.

**Evaluation on diffusion models:**
I also appreciated that the authors evaluated their attack on diffusion models, broadening the scope of their work.

**Removal Operation:**
The authors assess their method with different removal operations, as shown in Table 3. I found this information very valuable. One possible addition would be a discussion on why certain methods perform better than others.

**Weaknesses:**

**Suggestions for Improving the Paper:**

1. **Reframing of Existing Attack by Adding Noise to Input**
   The core idea of this paper is that points most vulnerable to membership inference attacks (MIAs) lie in sensitive regions of the feature space. These sensitive features, when perturbed, change the model's output. However, this approach appears to be a reframe of an earlier work [1], with the key difference being that [1] adds noise across the entire input, whereas this paper focuses on specific parts. This alone may not constitute a sufficient contribution. The authors could strengthen this work by offering fresh insights into the role of feature sensitivity. For instance, do sensitive features in images vulnerable to MIA exhibit identifiable patterns?

2. **Insufficient Information for Reproducing Results**
   Several details are missing, making reproduction challenging:
   - How were the shadow models trained, including parameters?
   - How many shadow models and target models were used?
   - How often were experiments repeated?
   - What are the train-test accuracies of these models?
   - What thresholds are used to separate members from non-members?

3. **Potential Overuse of CW for Identifying Vulnerable Features**
   Could simpler methods, such as super-pixels or saliency maps, be considered instead? Using CW could be costly, as it adds significant operations for attackers.

4. **Lack of Comparison with Stronger, Recent Attacks**
   The authors did not evaluate newer, stronger attacks, particularly [2,3], which demonstrate significant improvements over existing approaches. The only powerful attack compared here is LiRA, and even then, their method performs similarly (Fig. 4). Moreover, [2,3] can significantly outperform LiRA. To strengthen this work, the authors should include comparisons against these methods.

5. **Lower AUC Despite High TPR at 0.1% FPR**
   Table 1 shows results across various attacks. While the authors’ method outperforms existing attacks at low FPR, the AUC remains lower than others. Could this discrepancy be explained? Additionally, what are the standard deviations for the values in the table?

6. **Clarity in Figures**
   The TPR vs. FPR figures could benefit from improved clarity. For example, in Fig. 4, distinguishing between LiRA and the current attack is difficult. Could the authors provide AUCs and TPR@0.1% FPR for clarity?

7. **Absence of Requirement for Shadow Models**
   A major strength of this attack is its ability to be executed on pre-trained models without requiring shadow model training from scratch, unlike LiRA and [2]. Currently, the paper suggests that a shadow model is required. To highlight this advantage, the authors could restructure the paper to emphasize this point, as it significantly enhances the work’s value.

---

**References:**
[1] Bargav Jayaraman, Lingxiao Wang, David Evans, and Quanquan Gu. Revisiting membership inference under realistic assumptions. *Proceedings on Privacy Enhancing Technologies (PoPETs)*, 2021.
[2] *Low-Cost High-Power Membership Inference Attacks*
[3] Ye, J., Maddi, A., Murakonda, S. K., Bindschaedler, V., and Shokri, R. *Enhanced Membership Inference Attacks Against Machine Learning Models.* In *Proceedings of the 29th ACM SIGSAC Conference on Computer and Communications Security (CCS'22)*, pp. 3093–3106, 2022.

**Questions:**

See weaknesses.

---

> ### Author Response · Authors · 2024-11-20
> **Response to Reviewer NyRP (1/3)**
>
> Thank you for your detailed feedback, we greatly appreciate your positive remarks and constructive suggestions. Below, we address each of your concerns point by point.
>
> **Q1: Reframing of Existing Attack by Adding Noise to Input**
>
> Thank you for your insightful comments. While both our work and [1] utilize feature perturbation in the context of membership inference attacks, there are several key differences that set our approach apart:
>
> **Targeted Feature Perturbation vs. Indiscriminate Noise Addition**: Our approach emphasizes the importance of individual features and their specific contributions to the model's decision-making process. Not all regions of an image equally influence the model's output, as extensively studied in explainable machine learning research [r1, r2]. For example, saliency regions containing distinctive object parts or textures may be more critical for the model’s predictions. In contrast, [1] adds random noise indiscriminately across the entire input, which may not fully exploit the feature discrepancies between member and non-member samples. We, on the other hand, progressively perturb individual features, and we train a model to predict which features should be perturbed first. This results in a more fine-grained and targeted strategy that better leverages the differences in feature sensitivities.
>
> **Attributing Prediction Change to Feature Perturbation**: The success of a perturbation-based attack heavily relies on ensuring that the perturbation operations accurately reflect the importance of the features being perturbed, so that the changes in the model's predictions can be directly attributable to the perturbation of those features. The approach in [1], which involves adding random noise, can introduce distribution shifts (Hooker et al., 2019) and adversarial artifacts (Fong & Vedaldi, 2017). These unintended changes can obscure whether the output variations are due to feature perturbation or other factors. In our work, we designed specific perturbation operations to minimize such impacts, ensuring that the perturbations more accurately represent feature importance.
>
> To empirically demonstrate the effectiveness of our approach compared to [1], we implemented the Merlin attack from [1] under the same experimental settings as our method. Our results show that our method achieves a TPR of 4.6% at a 0.1% FPR on ResNet-18 trained on CIFAR-100, whereas the Merlin attack achieves a TPR of only 1.3% under the same conditions. We have revised Section 2 to clarify the distinctions between our method and that of [1].
>
>
> **Q2: Insufficient Information for Reproducing Results**
>
> Due to space limitations, we have provided detailed experimental settings in Appendix A. We apologize for any confusion and will make these details clearer in the manuscript. We will also publicly release our code to facilitate the reproduction of our results. Here, we address your questions point by point:
>
> - The shadow model is trained on a randomly selected subset from the same distribution as the target model but disjoint from its training data. It is trained for 100 epochs with an initial learning rate of 0.1. We utilize a cosine annealing schedule to gradually reduce the learning rate, employ standard data augmentations, and apply a weight decay rate of 0.0001.
>
> - In the standard adversarial setting, we use one shadow model and one target model. For settings with large supplementary datasets, we follow the approach of Liu et al. (2022), applying knowledge distillation for both shadow and target models and storing intermediate model checkpoints. For settings with a large number of shadow models, we follow the strategy of Carlini et al. (2022) and employ 8, 16, and 64 shadow models alongside a single target model.
>
> - We have repeated the experiments 10 times with different random samplings of the datasets, except for the experiments based on Carlini et al. (2022) due to the large computational costs associated with training shadow models.
>
> - The training and testing accuracies of all models are provided in Table 7.
>
> - We have adopted the similar strategy discussed in [1-3], we use the softmax output of the attack model as membership scores, which indicates the attack model’s prediction confidence. We vary the threshold between 0 and 1 to calculate the TPR at each swept FPR, simulating the adversary’s power and error based on these membership scores.
>
> We have revised Section 4.1 and Appendix A.2 to clarify the unclear details.

---

> ### Author Response · Authors · 2024-11-20
> **Response to Reviewer NyRP (2/3)**
>
> **Q3: Potential Overuse of CW for Identifying Vulnerable Features**
>
> Thank you for pointing this out. We have compared with different super-pixel methods, and the results are discussed in Appendix C - Different Removal Strategy and Table 8. Our method significantly outperforms super-pixel methods by a large margin. Regarding saliency map methods, since we assume a black-box threat model for the target model, we cannot utilize methods that require white-box knowledge, such as Integrated Gradients [r1] and GradCAM [r2]. Therefore, we selected two widely used saliency methods that work in black-box settings: LIME [r3] and SHAP [r4]. The results are as follows:
>
> | Saliency Method | TPR at 0.1% FPR | Balanced Accuracy | AUC |
> |:----------:|:----------:|:----------:|:----------:|
> | LIME  | 4.2%  | 0.821  | 0.940 |
> | SHAP  | 4.0%   | 0.813   | 0.938 |
>
> While these methods only perform slightly lower than our approach, we want to highlight that these black-box saliency methods are generally very computationally expensive. They require a large number of queries to approximate feature importances—typically several hundred to a thousand queries per sample to derive a saliency map. In contrast, our method is very efficient: training a mask prediction model takes less than 2 minutes on an Nvidia 4090 GPU, and it does not require access to the target model to derive the mask.
>
>
> **Q4: Lack of Comparison with Stronger, Recent Attacks**
>
> Thank you for your valuable feedback. We have conducted additional experiments to compare our method with the newer, stronger attacks mentioned in [2] and [3]. Specifically, we reproduced these attacks using their official implementations on WideResNet trained on CIFAR-10 and compared them with our attack in the same settings. For the attack in [3], we selected Attack R, which achieves the highest TPR at low FPR. We trained 16 shadow models, and the experimental results with different their attack strategies are as follows:
>
> | Attack Method | TPR at 0.1% FPR | AUC |
> |:----------:|:----------:|:----------:|
> | logit_rescale  | 1.6%  | 0.636  |
> | linear_itp  | 1.2%   | 0.641   |
> | min_linear_logit  | 0.6%   | 0.646   |
> | mean_linear_logit  | 0.5%   | 0.649   |
> | Ours. Random.  | 2.6%   | 0.686   |
>
> For RMIA [2], which focuses on improving membership inference performance in low-cost settings, we followed their experimental setup by training 1, 2, and 4 shadow models. The results are as follows:
>
> | Method | Number of Shadow Models | TPR at 0.1% FPR | AUC   |
> |---------|:----------:|:----------:|:----------:|
> |         RMIA        | 1                        | 1.9%             | 0.658 |
> |        RMIA         | 2                        | 3.0%             | 0.668 |
> |         RMIA        | 4                        | 4.1%             | 0.673 |
> |        Ours. Random.         | 1                        | 2.5%             | 0.679 |
> |        Ours. Random.         | 2                        | 3.3%             | 0.689 |
> |        Ours. Random.         | 4                        | 4.4%             | 0.698 |
>
>
>
> As the tables indicate, our method outperforms both attacks from [2] and [3]. For the attack in [3], we improved the TPR at 0.1% FPR from 1.6% to 2.6%. Compared to SOTA attack RMIA [2], our method achieves higher TPR and AUC, especially when the number of shadow models is low. We have revised Section 4.2 to include these results.

---

> ### Author Response · Authors · 2024-11-20
> **Response to Reviewer NyRP (3/3)**
>
> **Q5: Lower AUC Despite High TPR at 0.1% FPR**
>
> We would like to clarify that Yeom et al., Shokri et al., Salem et al., Song & Mittal, Ours. Random. and Ours. Guided., were evaluated under the same standard adversarial settings. In contrast, Shi et al., Liu et al., and our method using loss trajectory (Ours w/ loss traj.) were tested in adversarial settings with access to a larger auxiliary dataset. Our method can outperform baselines in both settings in AUC, while Liu et al. and Shi et al. achieved a slightly higher AUC than Ours. Random and Ours. Guided, this is due to their use of an auxiliary dataset, which enhances their attack capability.
>
> We have included the mean and standard deviation values in the following table for ResNet-18 trained on CIFAR-100. These results demonstrate that our method consistently outperforms the baselines, and the differences are statistically significant.
>
>
> | Attack Method       | TPR at 0.1% FPR | Balanced Accuracy | AUC   |
> |:---------------------:|:-----------------:|:----------------:|:-----------------:|
> | Yeom et al.        | 0.3±0.0%        | 0.770±0.022       | 0.770±0.002  |
> | Shokri et al.      | 0.7±0.1%        | 0.692±0.015       | 0.751±0.004  |
> | Salem et al.       | 0.6±0.1%        | 0.712±0.024       | 0.765±0.003  |
> | Song & Mittal      | 0.3±0.0%        | 0.771±0.026       | 0.771±0.002  |
> | Ours. Random.      | 4.1±0.2%        | 0.810±0.030       | 0.938±0.006  |
> | Ours. Guided.      | 4.6±0.3%        | 0.825±0.026       | 0.946±0.003  |
> | Shi et al.         | 2.8±0.2%        | 0.873±0.020       | 0.935±0.003  |
> | Liu et al.         | 8.2±0.5%        | 0.854±0.014       | 0.931±0.007  |
> | Ours w/ loss traj. | 12.0±0.5%       | 0.882±0.016       | 0.952±0.003  |
>
>
> **Q6: Clarity in Figures**
>
> Sorry for the confusion. We have provided the TPR at 0.1% FPR and the AUC in the table below, showing that our method consistently outperforms LiRA. We have revised Section 4.2 to make it clearer.
>
> |           Method           | TPR at 0.1% FPR |   AUC   |
> |:--------------------------:|:----------------:|:-------:|
> |       LiRA (n=8)          |       1.8%       |  0.591  |
> |    Ours w/ LiRA (n=8)     |       3.2%       |  0.652  |
> |       LiRA (n=16)         |       2.7%       |  0.655  |
> |    Ours w/ LiRA (n=16)    |       3.8%       |  0.676  |
> |       LiRA (n=64)         |       7.7%       |  0.693  |
> |    Ours w/ LiRA (n=64)    |       8.4%       |  0.701  |
> |   LiRA Aug. (n=64)        |       8.1%       |  0.722  |
> | Ours w/ LiRA Aug. (n=64)  |       8.6%       |  0.728  |
>
>
> **Q7: Absence of Requirement for Shadow Models**
>
> Thank you for your insightful and constructive feedback. We agree that a significant strength of our attack is its ability to be executed on pre-trained models without requiring the training of shadow models. To validate this point, we conducted preliminary experiments following the same settings as the population attacks in [3], without training shadow models. We implemented both attacks on ResNet-18 trained with CIFAR-100. The population attack achieved a TPR of only 0.4% at a 0.1% FPR with an AUC of 0.771. In contrast, our method achieved a TPR of 1.9% and an AUC of 0.916 under the same settings, significantly improving upon the population attack. This demonstrates the advantage of our approach, which does not require shadow models and is particularly beneficial for large-scale models, thereby significantly reducing computational costs. We have revised Section 1 and put a detailed discussion in Appendix F to emphasize this point.
>
>
> **Reference:**
>
> [r1] Sundararajan, Mukund, Ankur Taly, and Qiqi Yan. "Axiomatic attribution for deep networks." International conference on machine learning. PMLR, 2017.
>
> [r2] Selvaraju, Ramprasaath R., et al. "Grad-cam: Visual explanations from deep networks via gradient-based localization." Proceedings of the IEEE international conference on computer vision. 2017.
>
> [r3] Ribeiro, Marco Tulio, Sameer Singh, and Carlos Guestrin. "" Why should i trust you?" Explaining the predictions of any classifier." Proceedings of the 22nd ACM SIGKDD international conference on knowledge discovery and data mining. 2016.
>
> [r4] Lundberg, Scott M., and Su-In Lee. "A Unified Approach to Interpreting Model Predictions." Advances in Neural Information Processing Systems 30 (2017).

---

> > ### Comment · Reviewer_NyRP · 2024-11-25
> >
> > Thank you for the changes.
> > I think you have addressed most of my concerns. One thing I mentioned in the main review was showing if the model is using a different set of features for normal point vs leaked one. For example, if its a picture of a dog that is leaked (i.e., vulnerable to MIA), is the saliency map focused on the background instead of the dog. If the answer is yes, and that the model is looking at the wrong parts of the image, I would appreciate it if the authors can knit that into the story. I am open to increasing my score if the authors can do such a simple study to answer this question.
> >
> > There are a number of attacks out there, and this experiment will really make the work standout.

---

> > > ### Author Response · Authors · 2024-11-25
> > > **Response to Reviewer NyRP**
> > >
> > > Thank you for your response and for the insightful question. We agree that investigating whether the model uses different sets of features for member samples vulnerable to MIA versus non-member samples is important and can significantly strengthen our paper.
> > >
> > > To address this, we conducted an experiment using a common set of samples from the CIFAR-100 dataset. In the first scenario, these samples were included in the training process, serving as member samples. In the second scenario, the same samples were excluded from training, making them non-member samples. We generated saliency maps for these samples in both scenarios using GradCAM and measured their similarity using the Structural Similarity Index Measure (SSIM). The SSIM value between the saliency maps of member and non-member samples was computed to be $0.51 \pm 0.17$. We provide illustrative examples in Figure 8, which reveal clear disparities between the saliency maps of member and non-member samples. Specifically, the saliency maps for member samples focus more prominently on key semantic features—namely, the main objects critical for classification. In contrast, the saliency maps for non-member samples either fail to concentrate on these critical features or only partially engage with them. These observations help explain the efficacy of our guidance-based removal approach. By estimating feature importance and removing the least important features, our method minimizes the likelihood of affecting critical features in member samples. Conversely, non-member samples, which exhibit a weaker focus on key features, are less robust to feature removal.
> > >
> > > We have updated Appendix H to include these new insights and their implications for our method's effectiveness. Please let us know if you have any further questions.

---

### Official Review · Reviewer_9GYb · 2024-11-01

**Soundness:** 2
**Presentation:** 2
**Contribution:** 2
**Rating:** 5
**Confidence:** 3

**Summary:**

This paper studies the membership inference attack for image models. With the intuition that members and non-members would have different behaviors when losing important features, this paper proposes an algorithm where the adversary is assumed to have a shadow dataset to create a tuple of model, members and non-members, and then a member-vs-non-member discriminator is learned given the label, the loss and the after-removal features. The empirical results look promising, which are higher than the popular methods in the literature.

**Strengths:**

1. To my best knowledge, the intuition in the method is quite novel by considering the after-removal features as an important feature for membership classification.
2. The method as my understanding is mild in terms of computational cost, which avoids to train another model similar to the target model.
3. The main result looks promising. As shown in Table 1, the proposed method have certain benefit at TPR at low FPR.

**Weaknesses:**

The writing and the clarity need to be further improved. In details
1. It is not clear what the connection is between the intuition and the exact method. In section 3.2, the design intuition is described by the feature embedding $\phi$ and the density. However, it is not mentioned anymore in the later method design.
2. The method is not described very clear.
- In section 3.4, the paper introduces how to optimize the mask $m$. However, it does not define the domain of the mask: whether it is discrete in {0, 1} or it is in any bounded continuous region.
- It is not defined how to shape the original sample $x$ by the learned mask above to get the final "removal-based membership features", which is a part of the membership features in line 194. What is the removal process.
- It is also questionable how an MLP would work if the "removal-based membership features" is still in the pixel space, as introduced in line 199-200.
3. The variants of methods in the experiment, Ours. random, Ours Guided, Ours w/ loss traj. are not well-defined. For example, the loss trajectory is firstly mentioned in the background section, and then appears in the experiment result description paragraph.

**Questions:**

Please check the weakness section.

---

> ### Author Response · Authors · 2024-11-20
> **Response to Reviewer 9GYb**
>
> Thank you for your constructive feedback. We provide detailed responses to your comments.
>
> **Q1: It is not clear what the connection is between the intuition and the exact method. In section 3.2, the design intuition is described by the feature embedding and the density. However, it is not mentioned anymore in the later method design.**
>
> Thank you for the feedback. Our method is directly informed by the design intuition described in Section 3.2, where we observed that member samples tend to reside in high-density regions of the learned feature space. This suggests that the model is likely to exhibit higher confidence in these samples, even when some features are removed. Building on this insight, we propose a novel membership inference attack centered around the feature removal scheme described in Section 3.4. The feature removal operations, strategies, and model behavior selection are all closely aligned with this intuition. For instance, the design of the feature removal operation aims to accurately capture the importance of the removed features, ensuring that the changes in the model's predictions can be directly attributed to the removal of those features. We have revised Section 3.3 and Section 3.4 to clarify this connection.
>
> **Q2: The method is not described very clear.**
>
> - *Clarification of the mask*: The mask has the same shape as the input image, with each element taking a continuous value between 0 and 1. This design choice ensures that the optimization regarding the mask is end-to-end differentiable throughout the optimization process.
>
> - *Clarification of the removal process*: We apologize for the confusion. We have described the removal process in the implementation details. After training the mask prediction model, we input the original sample $x$ and obtain the predicted mask. We then rank the predicted mask values from lowest to highest, then remove features of the sample $x$ whose corresponding mask value percentiles fall below the specified removal ratio. The resulting sample $x'$ is then fed into the target model. We compute the model’s output behavior (e.g., confidence score for classification models or posterior estimation error for diffusion models). The sequence of output behaviors across different removal ratios forms the "removal-based membership features."
>
> - *Clarification of the removal-based membership features*: As explained earlier, the "removal-based membership features" are a sequence of model output behaviors (e.g., confidence scores) across different removal ratios. Therefore, they are no longer in the pixel space but instead represent model responses. This allows us to effectively train an MLP to distinguish between the output behaviors of members and non-members.
>
> We have revised Sections 3.3 and 3.4 to make the description clearer.
>
>
> **Q3: The variants of methods in the experiment, Ours. random, Ours Guided, Ours w/ loss traj. are not well-defined. For example, the loss trajectory is firstly mentioned in the background section, and then appears in the experiment result description paragraph.**
>
> Thank you for the feedback. We clarify the definitions as follows: Ours. random refers to our random-based removal strategy, where a specified ratio of pixels from the input is randomly removed. Ours. guided denotes our guided-based removal strategy, which ranks the predicted mask values from the mask prediction model in ascending order and removes the features of $x$ whose corresponding mask value percentiles fall below the specified removal ratio. Ours w/ loss traj. indicates that we implemented our attack in the same adversarial setting as Liu et al. (2022) and then concatenated our random removal-based attack features with the loss trajectories obtained using the method from Liu et al. (2022). We include the following table to better illustrate these variants:
>
> | Variant | Feature Removal Strategy | Additional Features Used |
> |:----------:|:----------:|:----------:|
> | Ours. Random.  | Random-based  | None  |
> | Ours. Guided.  | Guided-based   | None   |
> | Ours w/ loss traj.  | Random-based  | Loss trajectory (Liu et al., 2022)  |

---

> ### Author Response · Authors · 2024-11-25
>
> Dear Reviewer 9GYb,
>
> Thank you once again for your thoughtful feedback. We have provided a detailed response to your questions and hope it sufficiently addresses your concerns. With the interactive discussion period closing soon, please don't hesitate to reach out if you need any additional information or clarification.

---

> ### Comment · Reviewer_9GYb · 2024-11-27
>
> Dear authors,
>
> Thanks for your response and revised text for the method description. Your response addressed some of my clarity issues. With my (relatively) deeper understanding, I have further questions/concerns:
> 1. Why is it preferred to learn the U-net for predicting the  mask? An alternative way is to optimize $m$ for every $x$ w.r.t. the proposed loss function. Moreover, the mask $m$ is learned w.r.t the shadow model from my understanding, what's the intuition that it will still have the expected performance when it is applying with the samples in MIA test?
> 2. I am still not very convinced that the intuition is aligned with the method design. The intuition is discussing the learned feature space, where "learned" is in the sense of the feature representation (embedding space) learned with the training data.  However the "feature removal" in the method section is more about the "pixel removal" and it seems there is nothing related to the learned feature space. Also this introduces some confusions that "feature" in the intuition section means the embedding in the feature space, but later on "feature removal" seems to be the "pixel removal".
> 3. Clarity question: "final membership features" mentioned at line 220 of the revised pdf only has very few dimensions, which one dimensional confidence score of pixel removed image, the loss of original image, and the one-hot label + (potentially other signals from the literature)?

---

> ### Author Response · Authors · 2024-11-27
> **Response to Reviewer 9GYb**
>
> Thank you for your insightful feedback. We provide detailed responses to your additional questions.
>
> **Answer to Q1:**
>
> This is a good question. The preference for learning the U-Net model to predict the mask stems from several considerations. First, under the black-box threat model assumed in this work, the adversary does not have access to gradient information from the target model. As a result, directly optimizing the mask $m$ using the proposed loss function is not feasible because gradient-based updates cannot be performed. Additionally, while it is possible to use a shadow model to estimate the mask, optimizing $m$ for each sample individually would be computationally expensive. This process involves multiple components of the proposed loss function and requires numerous backpropagation steps for every sample. By contrast, training a U-Net model allows for efficient mask generation. After training, the U-Net produces the mask in a single forward pass, significantly improving the computational efficiency of the membership inference attack. Also, the U-Net model is particularly well-suited for this task due to its established effectiveness in image generation, e.g., in DDPM (Ho et al., 2020). U-Net utilizes feature maps at multiple resolutions to generate sharp and precise outputs.
>
> As for the second part of the question, our approach follows the threat models commonly adopted in prior works (Song & Mittal, 2021; Carlini et al., 2022; Liu et al., 2024). These threat models assume that the adversary is aware of the target model’s architecture and has access to an auxiliary dataset that shares the same distribution as the target model’s training data. Under this assumption, the shadow model, trained on the auxiliary dataset, is designed to closely replicate the behavior of the target model.
> The mask prediction model, trained using the outputs of the shadow model for both member and non-member samples, leverages this similarity. As the shadow model and target model exhibit comparable decision-making patterns, the mask prediction model is expected to generalize effectively to predict masks for samples from the target model.
>
> We have revised Appendix B to make the rationale clearer.
>
> **Answer to Q2:**
>
> Thank you for highlighting this concern. The distinction between feature removal and pixel removal arises due to the constraints imposed by the threat model. In our approach, pixel removal serves as a proxy to achieve the goal of feature removal.
>
> A neural network embeds an input sample into an intermediate feature representation, which is then used for subsequent classification. However, since we operate under a black-box threat model, the adversary cannot directly manipulate the intermediate feature representation. Instead, the adversary, having control over the input, can leverage pixel removal to approximate the feature removal process. For example, consider a neural network designed for image classification. The network embeds input samples into high-level feature representations, such as edges, textures, and shapes. When pixels are removed from the input (e.g., obscuring part of an object in an image), the corresponding high-level features, such as the edge structure, are also affected within the feature space. By simulating feature removal in this way, the adversary effectively influences the learned feature space, enabling the attack within the constraints of the threat model.
>
> We have revised Section 3.2 to make it clearer.
>
> **Answer to Q3:**
>
> The dimensionality of the final membership features depends on the dataset and the specific adversarial knowledge included. The confidence score of the pixel-removed image has a fixed dimensionality of 50, corresponding to 50 steps of pixel removal ratios (with fewer steps also explored in the ablation study). The one-hot label contributes 100 dimensions for CIFAR-100 and 10 dimensions for CIFAR-10 and CINIC-10. Additionally, when incorporating additional adversarial knowledge, such as methods from Liu et al. (2022a), loss trajectories contribute another 100 dimensions. The combination of these components and losses determines the size of the final membership features, which can become quite large depending on the configuration.
>
> Please let us know if you have any further questions.

---

### Official Review · Reviewer_iNv6 · 2024-11-03

**Soundness:** 2
**Presentation:** 2
**Contribution:** 3
**Rating:** 5
**Confidence:** 4

**Summary:**

This work proposes a new and efficient membership inference attack by exploiting the confidence change trajectory under varying feature removal. Its effectiveness is validated across both classification and generative models, applied to image and tabular data modalities.

**Strengths:**

S1: This work investigates a novel membership inference signal --- confidence change trajectory --- that requires fewer resources and lower computational costs than previous trajectory-based MIAs that often rely on additional model training.

S2: In addition to classification models, this MIA also supports diffusion models, demonstrating superior attack performance compared to existing state-of-the-art MIAs.

S3: The authors provide a thorough analysis of the feature removal strategy, highlighting the advantages of noisy linear imputation.

**Weaknesses:**

W1: Unclear intuition. In Section 3.2, the authors present two hypotheses: H1) member samples exhibit higher density values, and H2) member samples’ confidences should be more resilient, with H2 relying on H1. However, no supporting reference or empirical evidence is provided for H1. Additionally, Figure 1 does not effectively demonstrate H2, as the confidence changes for member samples lack consistency, e.g., the confidence drop for members is greater on WideResNet but smaller on ResNet compared to non-members.

W2: Incomplete experiments. Although the authors repeatedly highlight the resource efficiency of their MIA, there is no explicit comparison of resource or computational costs with state-of-the-art methods in the experimental section. Additionally, the work overlooks the resource requirements of the mask prediction model used.

W3: Unclear definition. The authors refer to random-based and guided-based removal strategies but do not provide clear definitions for these terms in the manuscript.

W4: Confusing terms. Both the title and the design intuition (Line 134) use the term “enhance” to describe this new MIA. For passive attacks, the gap between members and non-members remains relatively constant, with each signal having a different capacity to represent this gap. An effective membership signal can capture a clear and easily classifiable gap but does not actually increase it.

W5: Some presentation issues. For example, 1) Figure 2 is too small, making it difficult to distinguish each step clearly. 2) The term “the system” in Line 222 is unclear.

**Questions:**

Please refer to the Weaknesses section.

---

> ### Author Response · Authors · 2024-11-20
> **Response to Reviewer iNv6 (1/2)**
>
> Thank you for your constructive feedback. We provide detailed responses to your comments.
>
> **W1: Unclear Intuition.**
>
> **Regarding Hypothesis 1 (H1)**:
>
> Hypothesis 1 (member samples exhibit higher density values in the feature space) is inspired by feature learning theory (Cao et al., 2022; Kou et al., 2023). This theory suggests that a model's generalization capability is directly related to the training sample size and the signal-to-noise ratio of the features in the data. Extending this to membership inference, we introduce the concept of feature density, defined as the proximity of a sample's feature embedding to its nearest neighbors in the feature space.  We posit that member samples have higher density values in the feature space. To empirically validate this, we conducted experiments by extracting intermediate features from the layer immediately before the final classification layer for both member and non-member samples, then calculated the average L2 distance of each sample to its top 5 nearest neighbors in the feature space. The experiments were repeated across different models and datasets, and the mean and standard deviation of L2 distance is as follows:
>
> - ResNet-18 trained on CIFAR-100: Members: 71.11±1.08, non-members: 78.86±0.97
>
> - ResNet-18 trained on CIFAR-10: Members: 13.42±0.22, non-members: 15.52±0.34
>
> - WideResNet trained on CIFAR-100: Members: 54.34±0.78, non-members: 60.65±0.66
>
> - WideResNet trained on CIFAR-10: Members: 2.99±0.04, non-members: 3.22±0.03
>
> These results consistently indicate that member samples have lower mean L2 distances to their nearest neighbors compared to non-member samples, reflecting a higher feature density.
>
> **Regarding Hypothesis 2 (H2)**:
>
> Confidence resilience naturally arises from increased feature density, as redundant representations in high-density regions enable member samples to maintain higher confidence during the early stages of feature removal. Figure 1 shows a gradual decline in confidence scores for members compared to non-members in the early stage. We acknowledge that architectural differences (e.g., ResNet vs. WideResNet) influence the rate of these changes, however, despite variations in the rate of change, the overall trend remains consistent across architectures: a gradual decline in confidence scores for members, followed by a sharp drop in later stages. The sharp drop observed in the latter stage arises from significant feature removal, which shifts member embeddings from high-density regions to sparser areas in the feature space. This increased distance from the original embeddings causes a substantial drop in confidence scores. In contrast, non-member samples, which typically begin in lower-density regions with more uniformly distributed distances, experience a relatively steady decline in confidence throughout the feature removal process.
>
> We have clarified this design intuition in the revised manuscript, as reflected in Section 3.2 and Appendix D.
>
> **W2: Incomplete Experiments.**
>
> Thank you for highlighting this point. Resource efficiency is a key advantage of our approach, and we provide explicit comparisons to baseline methods to clarify this.
>
> **Comparison of Computational Costs:**
> The table below compares the computational resource requirements (number of trained models and training time) for our method and state-of-the-art methods, which demonstrates that our method could achieve better computational efficiency compared to previous works.
>
> | Attack Method | Number of Trained Models | Training Time |
> |:----------:|:----------:|:----------:|
> | Shi et al.  | 2  | 47.2 min  |
> | Liu et al.  | 2   | 51.7 min   |
> | Carlini et al.  | 64   | 1499.7 min   |
> | Ours. Random.  | 1   | 23.4 min   |
> | Ours. Guided.  | 2   | 24.6 min   |
>
> **Resource Costs for Mask Prediction Model:**
> Regarding the computational costs of the mask prediction model, it introduces minimal additional overhead. Training the mask prediction model takes less than 2 minutes on an Nvidia 4090 GPU, as it only requires 5 epochs to achieve high prediction performance.
>
> Furthermore, beyond computational resources, our method is also resource-efficient in terms of auxiliary data requirements. Many existing methods (e.g., Liu et al., 2022; Shi et al., 2024) require large auxiliary datasets with the same distribution as the target model's training data, in addition to shadow datasets. This requirement is often impractical in real-world scenarios. In contrast, our method does not rely on such large auxiliary datasets. We have revised Appendix E to include a comparison of attack costs.

---

> ### Author Response · Authors · 2024-11-20
> **Response to Reviewer iNv6 (2/2)**
>
> **W3: Unclear Definition.**
>
> We appreciate your feedback. The random-based removal strategy refers to randomly removing a specified ratio of pixels from the input. The guided-based removal strategy ranks the predicted mask values from the mask prediction model in ascending order and removes features of input whose mask values fall below a given percentile threshold, based on the specified removal ratio. We have revised Section 3.3 to clarify these definitions.
>
> **W4: Confusing Terms.**
>
> Thank you for pointing this out. As you noted, our approach involves proposing a new membership inference signal that more effectively captures the existing gap between members and non-members, rather than increasing that gap. We’ve adjusted our claim in Section 3.2 accordingly to reflect this more accurately.
>
> **W5: Presentation Issues.**
>
> Thank you for pointing this out. (1) We have enlarged Figure 2 to ensure that each step is distinguishable. (2) The term "the system" refers to the linear equation system. Specifically, the noisy linear imputation method approximates the values of removed pixels using a weighted mean of their neighboring pixels. When multiple pixels are removed, this process forms a system of equations where known pixel values are used directly, while removed pixels are treated as unknown variables, resulting in a linear equation system. We have revised Section 3.4 to make it clearer.

---

> ### Author Response · Authors · 2024-11-25
>
> Dear Reviewer iNv6,
>
> Thank you once again for your thoughtful feedback. We have provided a detailed response to your questions and hope it sufficiently addresses your concerns. With the interactive discussion period closing soon, please don't hesitate to reach out if you need any additional information or clarification.

---

> > ### Comment · Reviewer_iNv6 · 2024-11-26
> >
> > I appreciate the authors' efforts in the rebuttal. Regarding your response to W1, it appears that H2 is not much related to H1, as the distinct overall trends between members and non-members are sufficient for MIAs. Additionally, there seems to be an inconsistency in the rationale of H2, as it is discussed theoretically in the original manuscript but only shown empirically in the response. At this stage, I would maintain my original score.

---

> > > ### Author Response · Authors · 2024-11-26
> > > **Response to Reviewer iNv6**
> > >
> > > Thank you for your thoughtful feedback and additional questions. We would like to clarify the relationship between H1 and H2, which are inherently connected. H1 establishes the analysis framework that directly supports and motivates H2. Specifically, H1 demonstrates that member samples exhibit higher density values in the feature space. This higher density characteristic is not merely correlational but causally leads to the confidence resiliency observed between members and non-members in H2. The mechanism behind this connection is that high-density regions enable redundant representations, which allow member samples to maintain higher confidence levels during the progressive stages of feature removal.
> > >
> > > As you correctly pointed out, we also provide empirical evidence to demonstrate H2 explicitly. We believe that both our theoretical analysis and empirical results together provide robust support for our design intuition. Please let us know if you need any further clarification on this matter. We appreciate your time and guidance.

---

### Author Response · Authors · 2024-11-20
**Summary of Changes**

We sincerely thank all the reviewers for their valuable feedback and constructive suggestions!

In response to your insights, we have made revisions to improve our manuscript. Below is a summary of the key changes:

- [Reviewer iNv6] Adding empirical evidence and clarifying the design intuition: Section 3.2, Appendix D.

- [Reviewer iNv6, dYkK] Adding a comparison of resource and computational costs with state-of-the-art methods: Appendix E.

- [Reviewer iNv6, 9GYb] Adding clarifications on the definitions of different removal strategies: Section 3.3

- [Reviewer 9GYb] Adding a detailed description of the method design: Section 3.3, 3.4

- [Reviewer NyRP] Adding a detailed comparison with existing attacks of adding noises to input: Section 2

- [Reviewer NyRP] Added details on implementations and experimental settings: Section 4.1 and Appendix A.2

- [Reviewer NyRP] Adding comparison with different super-pixel and saliency map methods: Appendix C

- [Reviewer NyRP, dYkK] Adding comparison with stronger and recent attacks: Section 4.2

- [Reviewer NyRP, dYkK] Adding a metric table to better clarify Figure 4: Section 4.2

- [Reviewer NyRP]: Adding a discussion highlighting our methods' strength in not requiring shadow models: Section 1, Appendix F

- [Reviewer dYkK]: Adding experiments where the defender uses a pre-trained model for target model training: Appendix G

We hope these revisions and our detailed responses address your concerns. We sincerely appreciate your guidance and support.

---

### Meta-Review · Area_Chair_1jK5 · 2024-12-22

**Metareview:**

This paper presents a new membership inference attack that exploits change in the model's confidence when removing input features. This signal is efficient to compute and is effective against different model classes, including image classifiers and diffusion models. In addition, reviewers generally agree that the attack is novel and the evaluation is thorough and sound.

However, reviewers also cited several weaknesses, including lack of intuition for the attack, unclear explanation, and overall technical contribution being subpar for a top-tier conference such as ICLR. Given these weaknesses and the lack of enthusiastic support from reviewers, AC recommends rejection but encourages the authors to resubmit to a future venue after addressing the above weaknesses.

**Additional Comments On Reviewer Discussion:**

Changes are summarized in the author comment in https://openreview.net/forum?id=Jq8NPYVxLW&noteId=QLzeszZvar.

---

### Decision · Program_Chairs · 2025-01-22

Reject